# The role of Mediator and Little Elongation Complex in transcription termination

Hidehisa Takahashi[1 ✉], Amol Ranjan[2], Shiyuan Chen[2], Hidefumi Suzuki[1], Mio Shibata[3], Tomonori Hirose[1], Hiroko Hirose[1], Kazunori Sasaki[1], Ryota Abe[1], Kai Chen[2], Yanfeng He[2], Ying Zhang[2], Ichigaku Takigawa[4], Tadasuke Tsukiyama[3], Masashi Watanabe[3], Satoshi Fujii[5], Midori Iida[5], Junichi Yamamoto[6], Yuki Yamaguchi[7], Yutaka Suzuki[8], Masaki Matsumoto[9,10], Keiichi I. Nakayama[9,10], Michael P. Washburn[2,11], Anita Saraf[2], Laurence Florens[2], Shigeo Sato[2], Chieri Tomomori-Sato[2], Ronald C. Conaway[2,12], Joan W. Conaway[2,12 ✉] & Shigetsugu Hatakeyama[3 ✉]

Mediator is a coregulatory complex that regulates transcription of Pol II-dependent genes. Previously, we showed that human Mediator subunit MED26 plays a role in the recruitment of Super Elongation Complex (SEC) or Little Elongation Complex (LEC) to regulate the expression of certain genes. MED26 plays a role in recruiting SEC to protein-coding genes including *c-myc* and LEC to small nuclear RNA (snRNA) genes. However, how MED26 engages SEC or LEC to regulate distinct genes is unclear. Here, we provide evidence that MED26 recruits LEC to modulate transcription termination of non-polyadenylated transcripts including snRNAs and mRNAs encoding replication-dependent histone (RDH) at Cajal bodies. Our findings indicate that LEC recruited by MED26 promotes efficient transcription termination by Pol II through interaction with CBC-ARS2 and NELF/DSIF, and promotes 3′ end processing by enhancing recruitment of Integrator or Heat Labile Factor to snRNA or RDH genes, respectively.

[1] Department of Molecular Biology, Yokohama City University Graduate School of Medical Science, Fukuura 3-9, Kanazawa-ku, Yokohama, Kanagawa 216-0004, Japan. [2] Stowers Institute for Medical Research, 1000E 50th Street, Kansas City, MO 64110, USA. [3] Department of Biochemistry, Faculty of Medicine and Graduate School of Medicine, Hokkaido University, Kita 15, Nishi 7, Kita-ku, Sapporo, Hokkaido 060-8638, Japan. [4] Graduate School of Information Science and Technology, Hokkaido University, Kita 14, Nishi 9, Kita-ku, Sapporo, Hokkaido 060-0814, Japan. [5] Department of Bioscience and Bioinformatics, Kyushu Institute of Technology, Iizuka, Fukuoka 820-8502, Japan. [6] Department of Nanoparticle Translational Research Tokyo Medical University, 6-7-1, Nishi-Shinjuku, Tokyo, Shinjuku-ku 160-0023, Japan. [7] Graduate School of Bioscience and Biotechnology, Tokyo Institute of Technology, Yokohama, Kanagawa 226-8501, Japan. [8] Laboratory of Systems Genomics, Department of Computational Biology and Medical Sciences, Graduate School of Frontier Sciences, The University of Tokyo, 5-1-5, Kashiwanoha, Kashiwa, Chiba 277-8562, Japan. [9] Department of Molecular and Cellular Biology, Medical Institute of Bioregulation, Kyushu University, 3-1-1 Maidashi, Higashi-ku, Fukuoka, Fukuoka 812-8582, Japan. [10] Division of Proteomics, Medical Institute of Bioregulation, Kyushu University, 3-1-1 Maidashi, Higashi-ku, Fukuoka, Fukuoka 812-8582, Japan. [11] Department of Pathology and Laboratory Medicine, University of Kansas Medical Center, Kansas City, KS 66160, USA. [12] Department of Biochemistry and Molecular Biology, University of Kansas Medical Center, Kansas City, KS 66160, USA. ✉email: hide0213@yokohama-cu.ac.jp; JLC@stowers.org; hatas@med.hokudai.ac.jp

Eukaryotic RNA polymerase II (Pol II) is responsible for transcription of protein-coding genes and noncoding RNA genes, including most small nuclear RNAs (snRNAs), small nucleolar RNAs (snoRNAs), microRNAs (miRNA), and enhancer RNAs (eRNAs)[1]. Pol II-dependent transcription proceeds through multiple steps, including transcription initiation, elongation, and termination. Termination is tightly coupled to 3′-end processing of Pol II transcripts and is an essential process in transcript maturation[2,3].

In higher eukaryotes, there are three types of 3′-end processing of transcripts[4]. Although most protein-coding and noncoding RNA genes produce polyadenylated messenger RNAs (mRNAs) after 3′-end processing, some genes, including snRNA and replication-dependent histone (RDH) genes, produce transcripts lacking poly(A) tails using a distinct 3′-end processing machinery[2]. 3′-End processing of snRNA and RDH transcripts depends on different sets of cis-elements and transcription termination and 3′-end processing factors[5].

The 3′ ends of RDH mRNAs are generated by endonucleolytic cleavage at a site downstream from a conserved stem-loop structure and upstream of a purine-rich histone downstream element (HDE)[6]. The stem loop is bound by stem-loop-binding protein (SLBP), while HDE is recognized by U7 small nuclear ribonucleoprotein (snRNP)[6,7]. SLBP and U7 snRNP help to recruit heat-labile factor (HLF), composed of Symplekin, the multisubunit cleavage and polyadenylation specificity factor (CPSF), FLICE-associated huge protein (FLASH), and the 64-kDa subunit of cleavage stimulation factor (CSTF64)[6,8,9]. The CPSF subunit CPSF3 (CPSF73) is an endonuclease responsible for 3′-end cleavage of RDH transcripts[10]. Knockdown of termination factors results in read-through past normal sites of termination to conserved polyadenylation signals (PASs) just downstream of RDH HDEs and synthesis of polyadenylated RDH mRNAs, raising the possibility that polyadenylation of RDH transcripts could provide a mechanism to prevent Pol II from reading through into regions downstream of the RDH genes. In addition, recent evidence indicates that polyadenylated RDH transcripts contribute to the expression of RDH genes outside of S phase or in terminally differentiated, non-proliferating cells[11,12].

snRNA 3′-end formation depends on a 3′-box element ~20 nt downstream from the mature 3′ ends of snRNAs[5,13]. Integrator binds to the 3′ box and mediates 3′-end processing of pre-snRNA near the 3′ box[5]. Conserved PAS sequences are not found downstream of most snRNA genes, suggesting that during snRNA transcription, termination defects do not always lead to polyadenylation[14].

Recently, it has been shown that cap-binding complex (CBC) binds to arsenic resistance protein 2 (ARS2) to form CBC-ARS2 (CBCA) and contributes to termination of both RDH and snRNA transcription[15,16], consistent with the idea that CBCA links capping and elongation by Pol II to facilitate transcript termination. Furthermore, the negative elongation factor (NELF) and DRB sensitivity-inducing factor (DSIF), which are essential for promoter-proximal pausing by Pol II, interact with CBC and SLBP or Integrator to facilitate termination and 3′-end processing RDH or snRNA transcripts, respectively[14,17–19]. This evidence suggests that pausing by elongating Pol II contributes to termination of relatively short transcripts and facilitates 3′-end processing of non-polyadenylated transcripts.

Mediator, an evolutionarily conserved coregulatory complex that relays regulatory signals between gene-specific transcription activators and the basal initiation machinery[20,21], can regulate Pol II during pre-initiation complex assembly, initiation, and elongation[20,22]. Previously, we showed that in higher eukaryotes, Mediator, through the N-terminal domain (NTD) of metazoan-specific Mediator subunit MED26, interacts with transcription

elongation complexes designated super elongation complexes (SECs) and little elongation complexes (LECs)[23,24]. Members of the eleven-nineteen lysine-rich leukemia (ELL; in humans, ELL, ELL2, or ELL3) and ELL-associated factor (EAF, in humans, EAF1 or EAF2) families of transcription elongation factors (TEFs)[25–27] are shared components of SECs and LECs[23,24,28–30]. SECs also include positive TEFb and mixed lineage leukemia fusion partners[29–31]. MED26 helps to recruit SEC to a number of genes including c-myc and hsp70[23], where it regulates post-initiation events including phosphorylation of Rpb1 CTD and transcription elongation[23,28–30]. LECs contain ELL and EAF family members, the interactor of little elongation complex ELL subunit 1 (ICE1), interactor of little elongation complex ELL subunit 2 (ICE2), and are associated with zinc-finger CCCH-type containing 8 (ZC3H8) and ubiquitin-specific peptidase-like 1 (USPL1)[31–33]. Intriguingly, MED26 helps to recruit LEC to a subset of snRNA and snoRNA genes[24], where it contributes to their optimal expression;[24,32] however, it is an unanswered question why different ELL/EAF-containing elongation complexes are used to regulate different classes of genes. In addition, it remains unclear why LEC-associated ELL/EAF, whose only known function is to enhance Pol II elongation, would be needed for optimal expression of very short transcripts such as snRNAs.

In this report, we present a combination of genomic, proteomic, and molecular genetic evidence that human Mediator subunit MED26 recruits LEC to regulate transcription termination at RDH and snRNA genes and 3′-processing of RDH and snRNA precursors into mature, non-polyadenylated transcripts. In addition, we observe that Mediator subunits, like components of LEC, NELF, and HLF[7,19,24,31,34], are enriched at Cajal bodies, which are thought to be sites not only for pre-mRNA splicing but also for transcription regulation of snRNA and RDH genes[7]. Based on our findings, we propose the model that (i) MED26 recruits LEC to both RDH and snRNA genes in Cajal bodies, then (ii) LEC binds to CBCA and NELF/DSIF to inhibit Pol II read-through past the normal termination sites on these genes, and finally (iii) LEC promotes the 3′-end processing of the RDH or snRNA genes by enhancing recruitment of HLF or Integrator, respectively.

## Results

**Aberrant 3′-end processing of RDH genes by MED26 depletion.** To identify genes regulated by MED26, we performed RNA-sequencing (RNA-seq) of polyA-selected libraries prepared from HEK293T cells transfected with either control or MED26 small interfering RNA (siRNA). We identified 74 genes that were upregulated by a log2 ratio of at least 1.5 (false positive discovery rates [FDR] ≤0.05) following knockdown of MED26 (Fig. 1a and Supplementary Data 1). Strikingly, of the 74 upregulated genes, 21 were RDH genes. In a second experiment, we performed RNA-seq of ribo-depleted libraries, which are depleted of ribosomal RNAs but contain both non-polyadenylated and polyadenylated transcripts. Unexpectedly, we found that the majority of RDH transcripts found to be upregulated by MED26 depletion in polyA-selected libraries were either unchanged or present at reduced levels in ribo-depleted libraries.

The observation that MED26 depletion led to an increase in the abundance of RDH transcripts only in polyA-selected RNA-seq libraries raised the possibility that loss of MED26 leads to increased production of polyadenylated forms of RDH transcripts, which form when normal 3′-end processing fails to occur, allowing Pol II to continue transcription until it reaches a downstream PAS (Fig. 1b). To determine whether MED26 knockdown leads to accumulation of longer transcripts that extend beyond the normal 3′-end processing site of non-

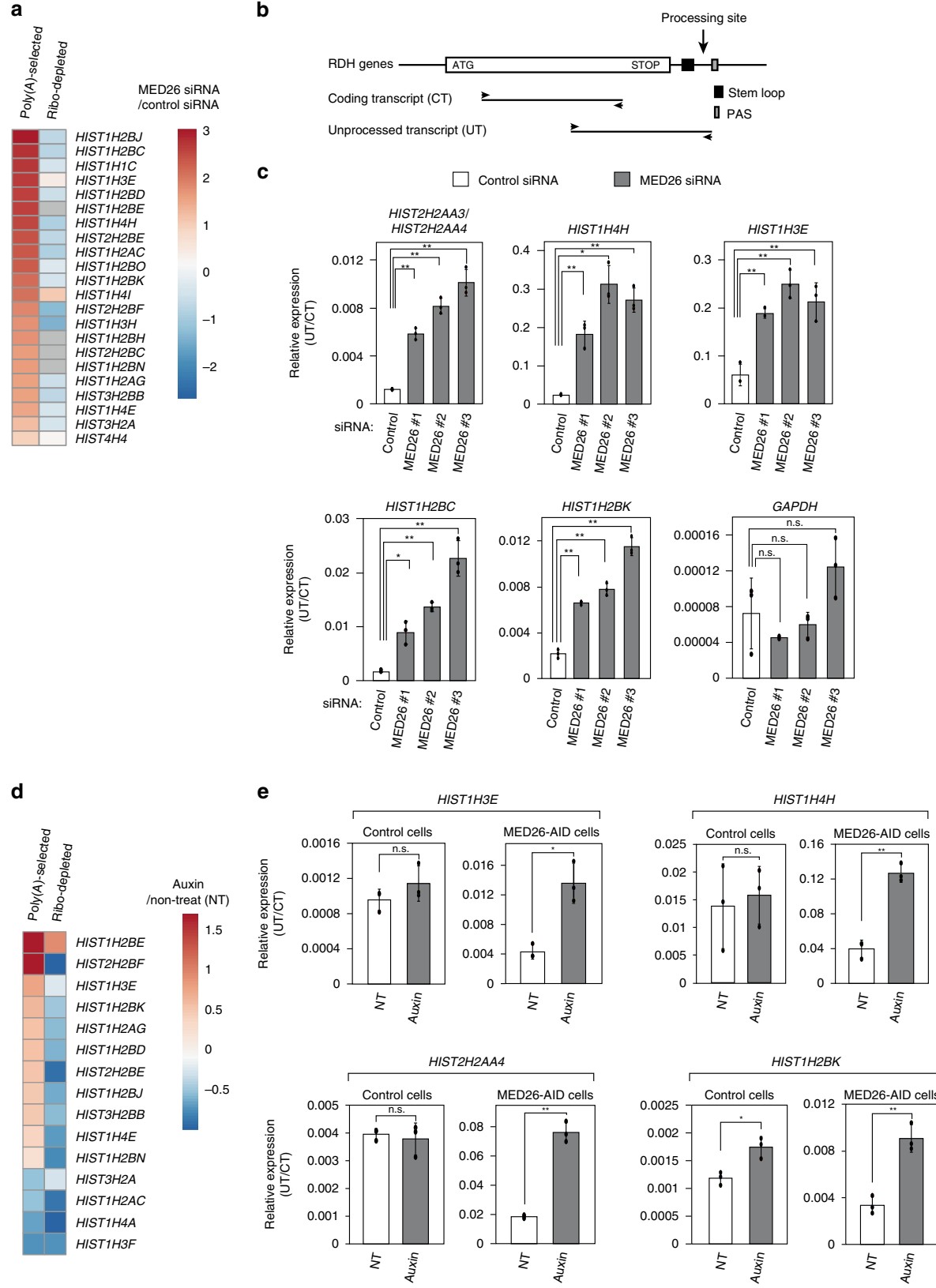

polyadenylated transcripts, we performed quantitative reverse transcription-PCR (RT-PCR) to measure levels of total RDH transcripts (coding transcripts, CTs) and unprocessed transcripts (UTs). The fraction of unprocessed RDH transcripts was significantly increased in cells transfected with each of three different MED26 siRNAs, while the fraction of unprocessed

glyceraldehyde 3-phosphate dehydrogenase (GAPDH) transcripts was unaffected (Fig. 1c). MED26 knockdown resulted in little change in levels of 3′-end processing and termination factors including SLBP, cap-binding protein 80 (CBP80), cap-binding protein 20 (CBP20), and U7 snRNA (Supplementary Fig. 1a, b), arguing against the possibility that the knockdown of MED26

**Fig. 1 MED26 depletion leads to aberrant 3′-end processing of RDH transcripts. a** Heat map showing differences in transcript abundance in HEK293T cells transfected with either control or MED26 siRNAs (log 2 siRNA MED26/control), determined by RNA-seq of polyA-selected or ribo-depleted libraries. **b** Organization of RDH genes and amplified regions of coding transcript and unprocessed transcript in quantitative real-time PCR analysis. 3′-End processing sites of RDH transcripts are localized between their stem-loop regions and downstream polyadenylation signals (PASs). **c** siRNA-mediated MED26 depletion increases the levels of unprocessed RDH transcripts. HEK293T cells were transfected with non-targeting siRNA as a control and each of three different siRNAs (#1, #2, and #3) targeting MED26. Total RNAs were extracted from cells and the ratio of unprocessed transcripts (UTs) to total transcripts (coding transcripts, CTs) from RDH genes was measured by real-time qPCR. Data points are the mean of three independent experiments, and error bars show standard deviation. The *P* values for the indicated comparisons were determined by Student's *t* test (*$P <$ 0.05; **$P < 0.01$). $n = 3$ biologically independent samples. Source data are provided as a Source Data file. **d** Heat map showing the differences in transcript abundance in parental HCT116 cells and HCT116 cells stably expressing MED26-AID in which the AID tag is fused to the C-terminus of MED26 and MED26 is acutely degraded in the presence of auxin. Transcripts derived from the cells were determined by RNA-seq of polyA-selected or ribo-depleted libraries (log2 auxin/non-treat). **e** Auxin-mediated MED26 degradation increased the levels of unprocessed RDH transcripts. Parental HCT116 cells or HCT116 cells stably expressing MED26-AID were cultured in the presence or absence of auxin. Total RNA was extracted from the cells and the ratio of UTs to total transcripts (CTs) from RDH genes was measured by real-time qPCR. Data points are the mean of three independent experiments and error bars show the standard deviation. The *P* values for the indicated comparisons were determined by Student's *t* test (*$P < 0.05$; **$P < 0.01$). $n = 3$ biologically independent samples. n.s., Not significant. Source data are provided as a Source Data file.

leads to aberrant 3′-end processing of RDH transcripts by affecting the expression of factors needed for efficient 3′-end processing and termination of RDH transcripts.

We performed similar experiments in an HCT116 cell line (HCT116-MED26-AID) expressing MED26 with a C-terminal auxin-inducible degron (MED26-AID), which is degraded upon addition of auxin to culture media (Supplementary Fig. 1c). RNA-seq using polyA-selected and ribo-depleted libraries suggested that polyadenylation of multiple RDH transcripts was increased after auxin treatment of HCT116-MED26-AID cells (Fig. 1d). In addition, auxin treatment increased the fraction of RDH transcripts that extend beyond the normal 3′-end processing site for non-polyadenylated transcripts in MED26-AID-expressing cells, but not in parental HCT116 cells (Fig. 1e).

**LEC contributes to efficient 3′-end processing of RDH genes.** To address the possibility that MED26 affects RDH transcript 3′-end processing and/or transcription termination through the recruitment of SEC or LEC, we performed quantitative RT-PCR to assay 3′-end processing efficiency in HEK293T cells transfected with siRNAs targeting MED26, LEC subunit ICE1, or SEC components AFF4 or CDK9. ICE1 and MED26 knockdown increased the fraction of unprocessed RDH transcripts, while knockdown of AFF4 or CDK9 did not (Fig. 2a and Supplementary Fig. 2a–d). In contrast, MED26 or ICE1 knockdown did not measurably change the fraction of UTs encoded by a known SEC target gene, *c-myc*, or by *GAPDH*. Knockdown of LEC-associated protein ZC3H8[31] (Supplementary Fig. 2e) also increased the levels RDH transcripts in polyA-selected libraries and of unprocessed RDH transcripts detected by quantitative PCR (qPCR) (Fig. 2b, c). We also performed RNA-seq of polyA-selected libraries from HEK293T cells transfected with either control, MED26, ICE1, or AFF4 siRNAs and identified 124 transcripts that were upregulated by the knockdown of either MED26 or ICE1, but not by AFF4 knockdown (Fig. 2d). These included many RDH transcripts (Fig. 2e and Supplementary Data 2), arguing that more polyadenylated RDH transcripts are generated following knockdown of either MED26 or ICE1. We note that many transcripts upregulated by ICE1 knockdown were not affected by MED26 knockdown. It was recently shown that ICE1 has a role in nonsense-mediated decay (NMD) of mRNAs outside the context of LEC[35]; thus, it is possible that ICE1 knockdown affects not only transcription of RDH and snRNA genes but also NMD, and perhaps other functions, at other genes. Taken together, our results support the idea that LEC, but not SEC, helps to regulate 3′-end processing and/or termination of RDH transcripts.

**MED26 contributes to LEC recruitment to RDH genes.** Since we previously showed that MED26 contributes to LEC recruitment to and regulation of snRNA genes[24], we considered the possibility that it also helps to recruit LEC to the RDH genes. Consistent with this idea, chromatin immunoprecipitation-sequencing (ChIP-seq) revealed that MED26 is present at both snRNA and RDH genes (Fig. 3a, b and Supplementary Data 3). To ask whether LEC is also present at MED26-occupied RDH genes, we compared our MED26 ChIP-seq dataset to published ChIP-seq datasets for ELL and the LEC-associated protein ZC3H8[31]. We identified 226 genes where MED26, ELL, and ZC3H8 co-occupied a region within 1 kb of the transcription start site (TSS) (Fig. 3c and Supplementary Data 3). Among these are both RDH and snRNA genes (Fig. 3c and Supplementary Fig. 3a–c). To address whether MED26 also plays a role in the recruitment of LEC to RDH genes, as well as snRNA genes, we tested whether the knockdown of MED26 affects the occupancy of LEC components at RDH genes. As shown in Fig. 3d, ELL occupancy at RDH genes is decreased following MED26 knockdown, as is occupancy of Pol II. Taken together, these and our previously published results[24] indicate that MED26 plays a role in the recruitment of LEC to both RDH and snRNA genes.

To address the effect of LEC recruitment by MED26 on the transcription regulation of the genes, we used a CRISPR (clustered regularly interspaced short palindromic repeats)-generated, MED26 hypomorphic mutant HEK293T cell line. This cell line expresses mutant MED26 that lacks the NTD required for LEC interaction with Mediator (Supplementary Fig. 4a)[24]. Based on semi-quantitative mass spectrometry of Mediator enriched via its ability to bind the transcriptional activation domain of ATF6α, we estimate that in these cells, mutant MED26 is present in Mediator at 30–50% of the level of its wild-type counterpart (Supplementary Fig. 4b). The results of RNA-seq experiments further indicated that the MED26 mutation gave rise to similar changes in gene expression to those observed after siRNA-mediated MED26 knockdown (Supplementary Fig. 4c). Furthermore, as shown in Supplementary Fig. 5a, the majority of the RDH transcripts upregulated by the MED26 mutation in polyA-selected libraries were decreased in ribo-depleted libraries. Consistent with these results, RT-qPCR analysis revealed that the fraction of unprocessed RDH transcripts was significantly increased in the MED26 hypomorphic mutant cells (Supplementary Fig. 5b). Because RDH genes are transcribed in the S phase of the cell cycle[6,7], we also examined the expression of RDH genes in the S phase in wild-type and MED26 hypomorphic mutant cells. We took advantage of aphidicolin (APH) to arrest the cell cycle at the S phase. As

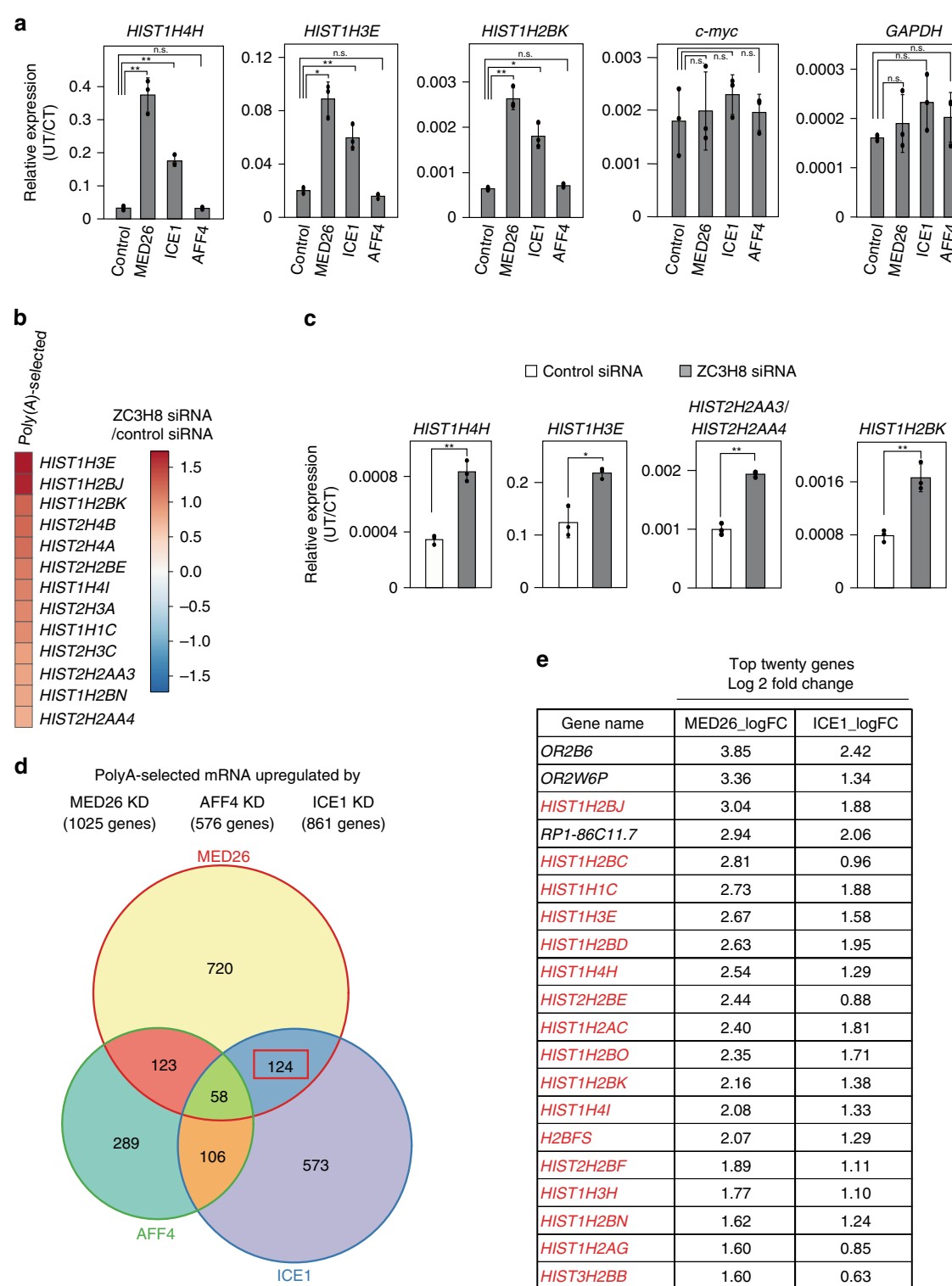

shown in Supplementary Fig. 5c, induction of RDH genes after the release of the cells from the S phase by APH removal was decreased in MED26 hypomorphic mutant cells, consistent with the ribo-depleted RNA-seq results showing that total levels of many RDH transcripts were decreased in MED26 hypomorphic mutant cells (Supplementary Fig. 5a). Taken together, our data suggest that MED26 has at least two functions in the synthesis of RDH transcripts: to promote optimal transcription of RDH genes and to increase the efficiency of RDH transcripts 3′ processing.

**RDH and snRNA gene termination defects in MED26 hypomorph.** Because defective 3′-end processing of RDH transcripts is often associated with defective transcription termination, we wished to determine whether MED26 contributes to proper termination at these and other genes. To do so, we performed Pol II ChIP-seq and precision nuclear run-on sequencing (PRO-seq) to compare the location of transcribing Pol II in wild-type and MED26 hypomorphic HEK293T cells. As shown in Fig. 4a–c and Supplementary Fig. 6a and 6c, inspection of browser tracks

**Fig. 2 LEC, but not SEC, is required for efficient 3′-end processing of RDH genes. a** Knockdown of MED26 and ICE1, but not AFF4, increased the ratio of unprocessed transcripts (UTs) to coding transcripts (CTs) of RDH genes, but not *c-myc* or *GAPDH* genes. Total RNAs were extracted from cells treated with control, MED26, ICE1, or AFF4 siRNAs, and unprocessed and coding regions were transfected and the ratio of UTs to CTs was measured by real-time qPCR. Data points are the mean of three independent experiments and error bars show the standard deviation. The *P* values for the indicated comparisons were determined by Student's *t* test (\*P < 0.05; \*\*P < 0.01). n = 3 biologically independent samples. Source data are provided as a Source Data file. **b** Heat map showing differences in transcript abundance in HEK293T cells transfected with either control or ZC3H8 siRNAs (log2 siRNA ZC3H8/control), determined by RNA-seq of polyA-selected libraries. **c** ZC3H8 knockdown increased the ratio of UTs and CTs of RDH genes. Data points are the average of three independent experiments and error bars show standard deviation. The *P* values for the indicated comparisons were determined by Student's *t* test (\*P < 0.05; \*\*P < 0.01). n = 3 biologically independent samples. Source data are provided as a Source Data file. **d** Venn diagram showing overlap of genes upregulated following knockdown of MED26, ICE1, and AFF4. The red box highlights a group of 124 genes increased by MED26 and ICE1 knockdown, but not by knockdown of AFF4. **e** The top 20 genes in groups affected by MED26 and ICE1 knockdown, but not by knockdown of AFF4, include many RDH genes. n.s., Not significant.

reveals an increase in Pol II ChIP-seq and PRO-seq signals downstream of multiple RDH genes in the mutant cells, suggesting that there is an increased level of read-through transcription past normal sites of transcription termination. Similarly, and consistent with our previous evidence that the MED26 NTD contributes to LEC recruitment to and proper expression of snRNA genes[24], Pol II ChIP-seq and PRO-seq signals were increased downstream of snRNA and snoRNA genes, suggesting that read-through transcription at these genes was also higher in MED26 mutant cells (Fig. 4d–f and Supplementary Fig. 6b, d). In contrast, we detected no increase in either Pol II or PRO-seq reads downstream of the normal cleavage and polyadenylation site at a well-characterized SEC target gene, *c-myc*, in the MED26 hypomorphic cells (Fig. 4g).

To address the possibility that the apparent increase in read-through transcription is simply a consequence of an increase in actively transcribing Pol II that reaches the 3′-end of the transcript coding sequence, we calculated a PRO-seq read-through ratio, defined as "the sum of reads from 500 to 1000 bp downstream of the 3′-end processing site (TES) divided by the sum of reads from TES to 50 bp upstream of TES." As shown in Fig. 4h (left panel) and Supplementary Fig. 6e, the PRO-seq read-through ratios of RDH genes and snRNA genes in mutant cells were significantly higher than those in wild-type cells. In contrast, the PRO-seq read-through ratios of other protein-coding genes in wild-type and mutant cells were similar. In addition, we calculated the Pol II read-through ratio defined as "the sum of Pol II reads from TES to 1000 bp downstream of TES divided by the sum of Pol II reads from TSS to 1000 bp downstream of TES." As shown in Fig. 4h (right panel), the Pol II read-through ratios of RDH genes and snRNA/snoRNA genes, but not other protein-coding genes, were significantly higher in mutant cells than in wild-type cells.

Of note, although we consistently observed increases in snRNA read-through transcripts in the MED26 mutant cells, following knockdown of MED26, we observed increased levels of just a few snRNA/snoRNA transcripts in our polyA-selected RNA-seq dataset (Supplementary Data 4), suggesting that, in contrast to RDH genes, the knockdown of MED26 caused only a little aberrant polyadenylation of snRNA/snoRNA genes. This result is consistent with the fact that conserved PAS sequences are not present at the downstream of most of snRNA genes in contrast to RDH genes[14]. Taken together, our results are consistent with the idea that MED26 plays a role in transcription termination of both RDH and snRNA genes. Although we cannot exclude the possibility that some of the observed effects were caused by a reduction in the absolute amount of MED26 in the mutant cells, our findings are consistent with the model that recruitment of LEC to RDH and snRNA genes via the MED26 NTD suppresses Pol II read-through and enhances normal transcription termination at both snRNA and RDH genes.

**LEC copurifies with CBCA, NELF/DSIF, HLF, and Integrator.** If MED26 plays a role in transcription termination through the recruitment of LEC to the genes, it seemed possible that LEC would interact with additional factors that contribute to transcription termination. To address this possibility, we generated 293FRT cell lines that stably express FLAG-tagged ICE1 and purified LEC through anti-FLAG affinity purification. We thought it was possible that functional LEC might be resistant to solubilization from nuclei, since we and others have shown that LEC and MED26 are colocalized at Cajal bodies, which are nuclear speckles composed of factors including coilin and are known to be the sites for transcription of snRNA and RDH genes[24,31,36]. We therefore extracted nuclear proteins in the presence of benzonase to digest nucleic acids, including both DNA and RNA, and purified LEC through anti-FLAG affinity purification (Fig. 5a). As shown in Fig. 5b, FLAG-ICE1 copurified with a number of proteins. To identify these proteins, we performed both multidimensional protein identification technology (MudPIT)-based proteomics and gel-based mass spectrometry analyses. As expected, ICE1 copurified with other LEC components, including large amounts of ICE2, ELL, and EAF1. It also copurified with smaller amounts of ELL2, EAF2, and ZC3H8, suggesting that (i) the majority of LEC in these cells is associated with ELL and EAF1 rather than ELL2 and EAF2 and (ii) ICE1, ICE2, and ELL/EAF1 are core components of LEC, while ZC3H8 is associated with only a subfraction of the LEC we have isolated (Fig. 5c). This observation, together with the fact that the number of genes associated with ZC3H8 was much greater than the number associated with ELL in ChIP-seq analyses (Fig. 3c), raises the possibility that ZC3H8 has functions outside LEC.

In addition to previously characterized LEC components, we identified Mediator subunits as LEC-interacting proteins. Although MED26 was not among the subunits detected in our mass spec datasets, it was readily detected by western blotting in FLAG-ICE1-immunopurified fractions (see Fig. 6a, b), consistent with our evidence that MED26-containing Mediator helps to recruit LEC to RDH and snRNA genes (Fig. 3d) and to regulate their transcription[24]. LEC also copurified with coilin and USPL1 (Fig. 5c), which are components of Cajal bodies, consistent with previous data showing that LEC and MED26 colocalize with Cajal bodies[24,31,37]. In addition, LEC copurified with nuclear protein co-activator of histone transcription, a marker of histone locus bodies (Fig. 5c). RDH gene loci are known to be present at histone locus bodies[7]. Furthermore, histone locus bodies are partly colocalized at Cajal bodies[6].

Finally, LEC copurified with a collection of proteins that have been implicated in termination and 3′-end processing of RDH and snRNA genes. Among these are components of HLF, including CPSF, CSTF, FLASH, and Symplekin; components of the Integrator, including INTS1, INTS4, INTS6, INTS7, and INTS12; components of the NELF, including NELFa, NELFb,

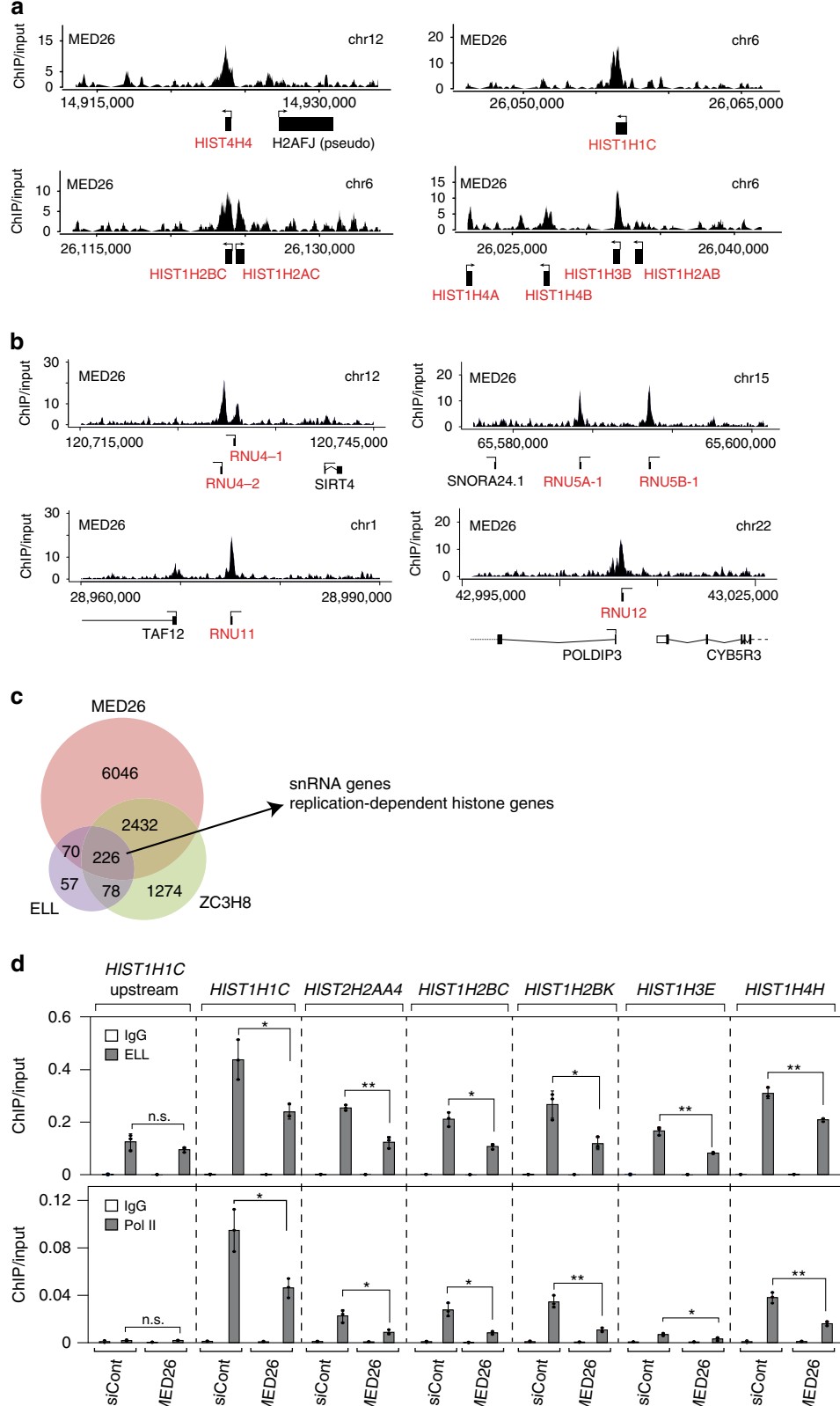

**Fig. 3 MED26 and LEC are present at RDH and snRNA genes. a, b** ChIP-sequence analysis was performed using duplicate ChIP samples. Examples of ChIP-seq genome browser tracks showing the presence of MED26 at RDH genes (**a**), and snRNA genes (**b**), are depicted. **c** Venn diagram showing the overlap of MED26-, ELL-, and ZC3H8-occupied genes. ELL and ZC3H8 ChIP-seq datasets are from are from GEO GSE47938[31]. **d** ELL and Pol II occupancy (ChIP/Input) at RDH genes is decreased by MED26 knockdown. Ct values of each ChIP are normalized to that of the input. Each value is the mean of three independent experiments and error bars show the standard deviation. The *P* values for the indicated comparisons were determined by Student's *t* test (*$P < 0.05$; **$P < 0.01$). $n = 3$ biologically independent samples. n.s., Not significant. Source data are provided as a Source Data file.

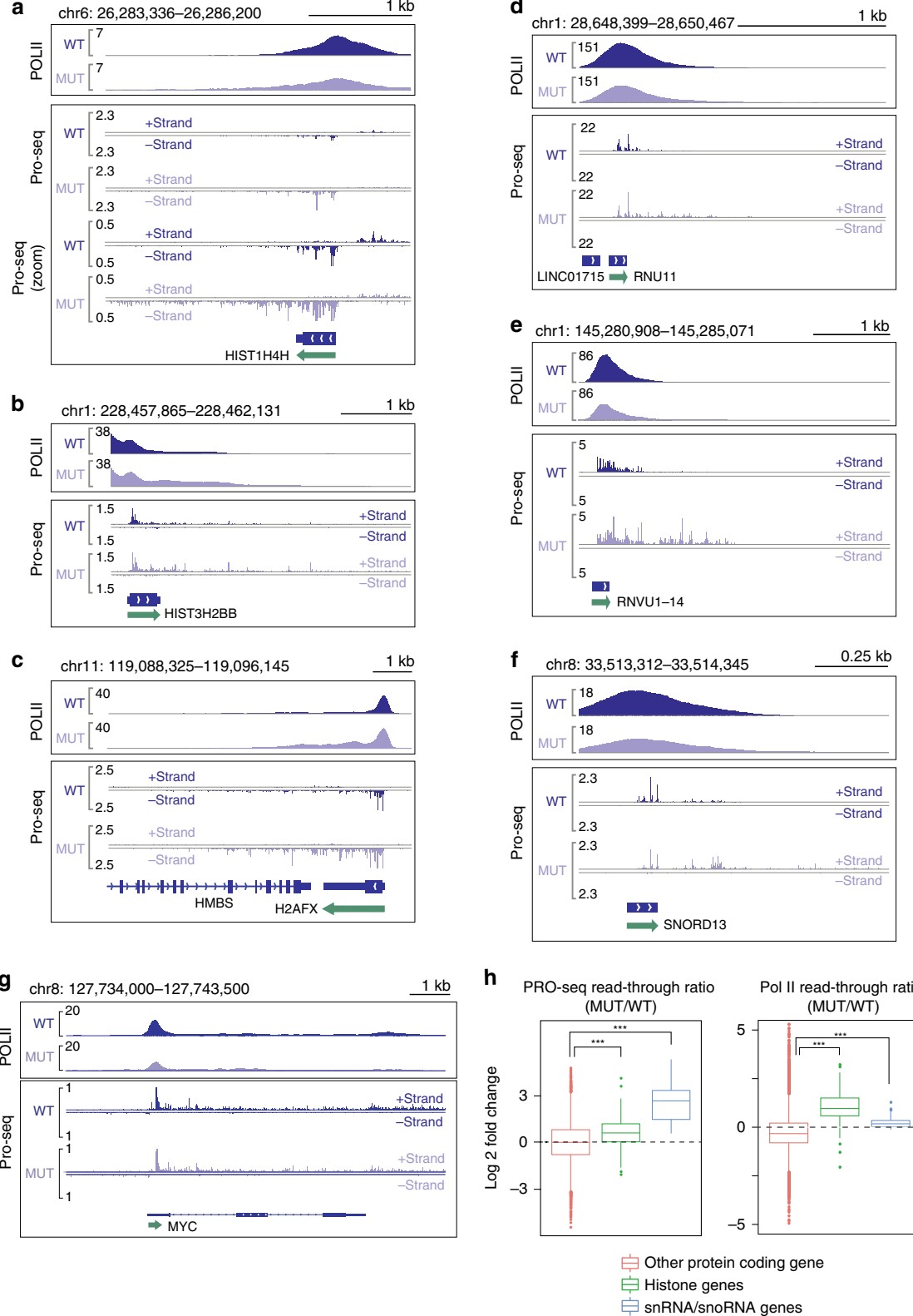

NELFc, NELFd, and NELFe; components of DSIF, including SPT5 and SPT6; and components of CBCA, including CBP80, CBP20, ARS2, and ZC3H18 (Fig. 5c).

CBCA plays an important role in transcription termination of both snRNA and RDH genes and inhibits Pol II read-through at these genes[15]. It was also shown that NELF/DSIF regulates transcription termination through interaction with CBC-SLBP or

Integrator and participates in the 3′-processing of RDH and snRNA genes, respectively[14,19]. It is thought that CBCA links the 5′ cap of nascent transcripts and elongating Pol II to inhibit Pol II read-through at snRNA and RDH genes. Our evidence that LEC specifically copurifies with transcription termination factors and 3′-end processing factors for RDH and snRNA genes is consistent with the model that LEC specifically regulates genes that encode

**Fig. 4 Increased read-through transcription at RDH genes and snRNA genes in MED26 hypomorphic mutant cells. a–c** Genome browser tracks showing the distribution of PRO-seq reads at RDH genes in wild-type (WT) HEK293T and MED26 mutant (MUT) cells. **d–f** Genome browser tracks showing the distribution of PRO-seq reads at snRNA genes in wild-type (WT) HEK293T and MED26 mutant (MUT) cells. **g** Genome browser tracks showing the distribution of PRO-seq reads at *c-myc* genes in wild-type (WT) HEK293T and MED26 mutant (MUT) cells. **h** Left panel shows box plot representing PRO-seq read-through ratios of histone genes ($n = 59$), snRNA/snoRNA genes ($n = 41$), and other protein-coding genes ($n = 8597$) in wild-type (WT) HEK293T and MED26 mutant (MUT) cells. Right panel shows Pol II read-through ratios of histone genes ($n = 68$), snRNA/snoRNA genes ($n = 40$), and other protein-coding genes ($n = 17809$) in wild-type (WT) HEK293T and MED26 mutant (MUT) cells. Center line of each box plot represents the median. Upper fence and lower fence of each box plot represent upper and lower quartiles, respectively. The range of each whisker represents 1.5 times the interquartile range. *P* values for the indicated comparisons were determined by Student's *t* test (***$P < 0.001$).

non-polyadenylated transcripts and raises the possibility that LEC inhibits Pol II read-through by interacting with CBCA, NELF/DSIF, and then recruits HLF and Integrator to facilitate 3′-end processing at RDH and snRNA genes, respectively.

To confirm and extend the results of our proteomic experiments, we used western blotting to show that FLAG-ICE1 copurifies with components of CBCA, NELF/DSIF, HLF, Integrator, and Mediator (Fig. 6a). In contrast, FLAG-tagged AF9, a component of SEC, did not copurify with these factors (Fig. 6a), consistent with our evidence that LEC, but not SEC, regulates transcription termination and 3′-end processing at genes encoding non-polyadenylated transcripts. As shown in Fig. 6a, b, FLAG-tagged ICE1 copurified with Mediator subunits, including MED26, MED1, and MED23, while a control protein, F-BTBD19, did not. Of note, FLAG-tagged AF9 also copurified with Mediator subunits, including MED26 and MED1 (Fig. 6a), consistent with previous evidence that MED26 interacts with SEC[23]. We also immunopurified endogenous ICE1 and associated proteins using two kinds of anti-ICE1 antibodies. As shown in Fig. 6c, both anti-ICE1 antibodies #1 and #2 immunoprecipitated NELFb, Symplekin, and INTS4, indicating that endogenous ICE1 interacts with the components of NELF, HLF, and Integrator. These results strongly support our proteomic results showing that LEC interacts with the components of CBCA-NELF/DSIF, HLF, Integrator, and Mediator.

Both FLAG-tagged ICE1 and FLAG-tagged ZC3H8 copurified with CBP80 (Supplementary Fig. 7), raising the possibility that CBC directly binds to LEC. Since we previously showed that the C-terminal region of ICE1 (CL: 1191-2266) binds directly to ICE2 and ELL and is sufficient for the formation of core LEC composed of ICE1, ICE2, ELL, and EAF, we considered the possibility that the N-terminal region of ICE1 (NL: 1-1190) might contribute to interaction with CBC. To test this possibility, we performed an in vitro binding assay using recombinant CBP80, CBP20, SLBP, and ICE1-NL proteins expressed in a baculoviral expression system. As shown in Fig. 6d, ICE1-NL directly bound to CBP80 in vitro, but not to CBP20 and SLBP.

**Mediator, LEC, NELF and HLF colocalize at Cajal bodies.** Cajal bodies are known to be sites not only for pre-mRNA splicing but also for transcription of snRNA genes[36,38]. RDH gene loci are known to be present at histone locus bodies[7]. As noted above, histone locus bodies are also partly colocalized at Cajal bodies, suggesting that transcription termination of non-polyadenylated snRNA and RDH transcripts is regulated in Cajal bodies[5,7]. Since we and others have previously shown that MED26 and components of LEC, NELF, and HLF were colocalized at Cajal bodies[7,19,24,32,38], we investigated whether other components of Mediator were also colocalized at Cajal bodies similarly to LEC, NELF, and HLF. We performed immunostaining of the Mediator components MED26, MED24, and MED6, the LEC component ICE1, the NELF component NELFb, the HLF component CSTF64, and a molecular marker of Cajal bodies, coilin. Consistent with previous results, ICE1, MED26, MED24, MED6,

NELFb, and CSTF64 were colocalized with coilin in Cajal bodies, but the SEC component AFF4 was not (Fig. 7a–g). Considering that Integrator has also been shown to be localized at Cajal bodies[34], our results suggest that Mediator, LEC, NELF, HLF, and Integrator have a role in transcription termination of non-polyadenylated genes, including snRNA and RDH genes at Cajal bodies.

To assess in more detail the nature of ICE1 and MED26-containing particles, we quantified the number of particles of ICE1 and MED26 colocalized or not colocalized with coilin and calculated their intensities. As shown in Fig. 7h, we found that 43 of 132 ICE1 particles identified in nuclei of 33 cells were colocalized with coilin at Cajal bodies and that the size of ICE1 particles colocalized with coilin was larger than that of another 89 ICE1 particles that did not colocalize with coilin. In addition, the signal intensity of ICE1 particles colocalized with coilin was much higher than that of particles not colocalized with coilin. Intriguingly, we observed the extranuclear regions of 33 cells and found that 130 ICE1 particles were present in the extranuclear area and that these particles had a much smaller size than particles colocalized with coilin in nuclei. Considering a recent report that ICE1 plays a role in NMD of mRNAs[35], it is possible that ICE1 has a role in NMD of mRNAs in the extranuclear region. In addition, we found that 59 of 176 MED26 particles in 18 nuclei were colocalized with coilin and that the size of MED26 particles colocalized with coilin was larger than that of another 117 MED26 particles not colocalized with coilin (Fig. 7i). In contrast to ICE1, we found just one MED26 particle in the extranuclear region, consistent with the known nuclear function of Mediator. In contrast to ICE1, the intensity of MED26 nuclear particles colocalized with coilin was similar to that of particles not colocalized with coilin. Although the identity of the MED26 particles not colocalized with coilin remains to be determined, it is tempting to speculate that they may be involved in transcriptional regulation of SEC-targeted genes.

To investigate the localization of Mediator and LEC in Cajal bodies at higher resolution, we observed MED26, MED24, MED6, ICE1, and coilin using stimulated emission depletion (STED) super-resolution microscopy[39]. Although the strength of staining for individual proteins was different, in Cajal bodies, Mediator subunits, ICE1, and coilin formed similar bead-like structures in which each bead appeared connected to other beads (Fig. 8a–d). These bead-like structures containing ICE1 or Mediator subunits and coilin were readily detected in line scan analyses across Cajal bodies (Fig. 8e–h). We also observed other speckles stained by the anti-MED26 antibodies adjacent to Cajal bodies (Fig. 8f), consistent with our result that MED26 was also present at locations other than Cajal bodies (Fig. 7b, i). Taken together, this result also supports our idea that Mediator and LEC play roles in transcription regulation of non-polyadenylated genes in Cajal bodies.

**LEC supports the recruitment of HLF or Integrator to genes.** The copurification of LEC with 3′-end processing factors for

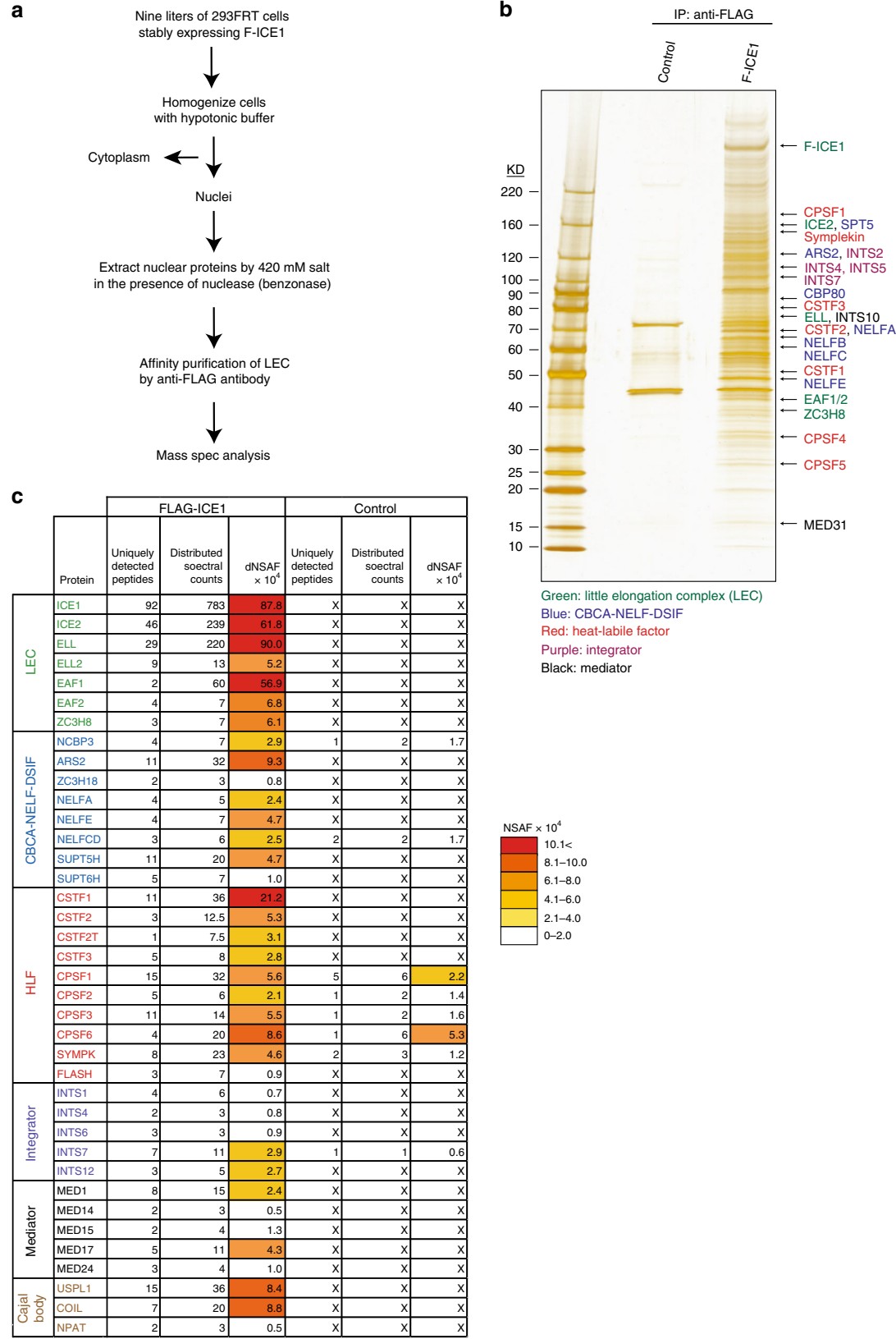

**a**
Nine liters of 293FRT cells stably expressing F-ICE1

↓

Homogenize cells with hypotonic buffer

Cytoplasm ← ↓

Nuclei

↓

Extract nuclear proteins by 420 mM salt in the presence of nuclease (benzonase)

↓

Affinity purification of LEC by anti-FLAG antibody

↓

Mass spec analysis

**b** IP: anti-FLAG

Green: little elongation complex (LEC)
Blue: CBCA-NELF-DSIF
Red: heat-labile factor
Purple: integrator
Black: mediator

**c**

| | Protein | FLAG-ICE1 | | | Control | | |
|---|---|---|---|---|---|---|---|
| | | Uniquely detected peptides | Distributed spectral counts | dNSAF × 10⁴ | Uniquely detected peptides | Distributed spectral counts | dNSAF × 10⁴ |
| LEC | ICE1 | 92 | 783 | 87.8 | X | X | X |
| | ICE2 | 46 | 239 | 61.8 | X | X | X |
| | ELL | 29 | 220 | 90.0 | X | X | X |
| | ELL2 | 9 | 13 | 5.2 | X | X | X |
| | EAF1 | 2 | 60 | 56.9 | X | X | X |
| | EAF2 | 4 | 7 | 6.8 | X | X | X |
| | ZC3H8 | 3 | 7 | 6.1 | X | X | X |
| CBCA-NELF-DSIF | NCBP3 | 4 | 7 | 2.9 | 1 | 2 | 1.7 |
| | ARS2 | 11 | 32 | 9.3 | X | X | X |
| | ZC3H18 | 2 | 3 | 0.8 | X | X | X |
| | NELFA | 4 | 5 | 2.4 | X | X | X |
| | NELFE | 4 | 7 | 4.7 | X | X | X |
| | NELFCD | 3 | 6 | 2.5 | 2 | 2 | 1.7 |
| | SUPT5H | 11 | 20 | 4.7 | X | X | X |
| | SUPT6H | 5 | 7 | 1.0 | X | X | X |
| HLF | CSTF1 | 11 | 36 | 21.2 | X | X | X |
| | CSTF2 | 3 | 12.5 | 5.3 | X | X | X |
| | CSTF2T | 1 | 7.5 | 3.1 | X | X | X |
| | CSTF3 | 5 | 8 | 2.8 | X | X | X |
| | CPSF1 | 15 | 32 | 5.6 | 5 | 6 | 2.2 |
| | CPSF2 | 5 | 6 | 2.1 | 1 | 2 | 1.4 |
| | CPSF3 | 11 | 14 | 5.5 | 1 | 2 | 1.6 |
| | CPSF6 | 4 | 20 | 8.6 | 1 | 6 | 5.3 |
| | SYMPK | 8 | 23 | 4.6 | 2 | 3 | 1.2 |
| | FLASH | 3 | 7 | 0.9 | X | X | X |
| Integrator | INTS1 | 4 | 6 | 0.7 | X | X | X |
| | INTS4 | 2 | 3 | 0.8 | X | X | X |
| | INTS6 | 3 | 3 | 0.9 | X | X | X |
| | INTS7 | 7 | 11 | 2.9 | 1 | 1 | 0.6 |
| | INTS12 | 3 | 5 | 2.7 | X | X | X |
| Mediator | MED1 | 8 | 15 | 2.4 | X | X | X |
| | MED14 | 2 | 3 | 0.5 | X | X | X |
| | MED15 | 2 | 4 | 1.3 | X | X | X |
| | MED17 | 5 | 11 | 4.3 | X | X | X |
| | MED24 | 3 | 4 | 1.0 | X | X | X |
| Cajal body | USPL1 | 15 | 36 | 8.4 | X | X | X |
| | COIL | 7 | 20 | 8.8 | X | X | X |
| | NPAT | 2 | 3 | 0.5 | X | X | X |

NSAF × 10⁴
10.1<
8.1–10.0
6.1–8.0
4.1–6.0
2.1–4.0
0–2.0

RDH and snRNA genes raised the possibility that LEC contributes to the recruitment of 3′-end processing factors to these genes. To examine this possibility, we investigated whether knockdown of ICE1 affected the occupancy of HLF at RDH genes and Integrator at snRNA genes. Because we found that the HLF component CPSF3 and Integrator component INTS9 copurified with FLAG-tagged ICE1 (Figs. 5c and 6a), we examined whether knockdown of ICE1 affected the occupancy of CPSF3 or INTS9 at RDH or snRNA genes, respectively. As shown in Fig. 9a, CPSF3 occupancy was decreased by knockdown of ICE1. Similarly, the

**Fig. 5 LEC copurifies with transcription termination factors including CBCA-NELF-DSIF, HLF, and Integrator. a** Schematic representation of the strategy for purifying LEC-binding proteins. Nuclear extract fractions from parental 293FRT cells and 293FRT cells stably expressing FLAG-tagged ICE1 were prepared in the presence of Benzonase® Nuclease. ICE1-binding proteins were purified from nuclear extracts through anti-FLAG affinity purification. **b** Silver staining of F-ICE1-binding proteins purified through anti-FLAG affinity chromatography. Each band of F-ICE1-binding proteins was cut out from the gels. Each gel was treated with trypsin and the resulting peptides from the gels were analyzed by mass spectrometry. **c** F-ICE1-binding proteins purified through anti-FLAG affinity chromatography or non-specific proteins present in negative FLAG control were identified through MudPIT (multidimensional protein identification technology)-based proteomic analysis. To estimate the relative protein levels, distributed normalized spectral abundance factors (dNSAF) were calculated for each detected protein[57,62–64].

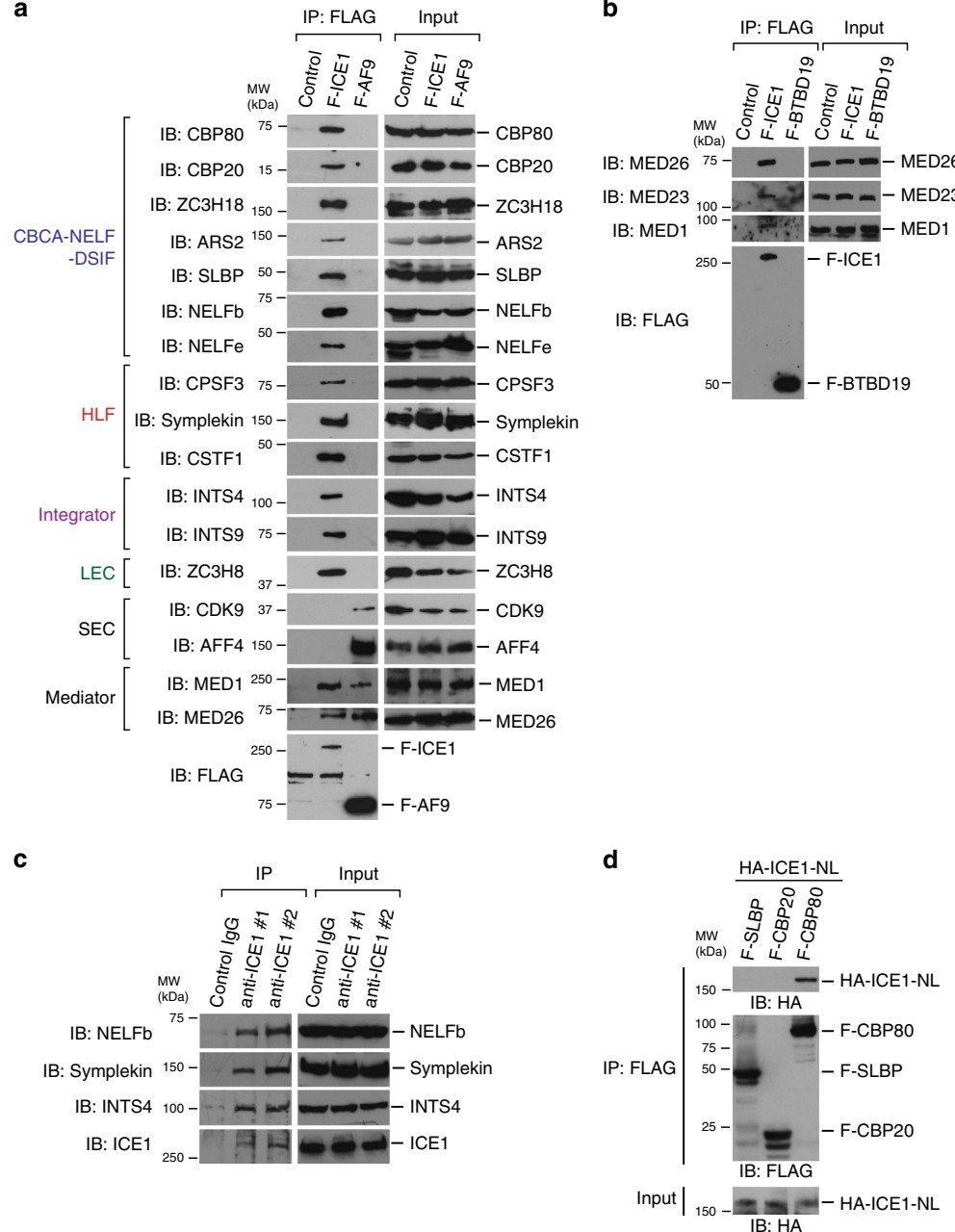

**Fig. 6 ICE1 copurified with transcription termination factors. a** Western blotting for FLAG-immunopurified complexes from parental 293FRT cells (control) and 293FRT cells stably expressing FLAG-tagged ICE1 and FLAG-tagged AF9. Source data are provided as a Source Data file. **b** Western blotting analysis of FLAG-immunopurified complexes from parental 293FRT cells (control) and 293FRT cells stably expressing FLAG-tagged ICE1 and FLAG-tagged BTBD19. Source data are provided as a Source Data file. **c** Western blotting analysis of control IgG, anti-ICE (#1), and anti-ICE1 (#2) immunopurified complexes from nuclear extracts of HEK293T cells. Source data are provided as a Source Data file. **d** In vitro binding assays using baculovirally expressed FLAG-tagged SLBP, CBP80, CBP20, and HA-tagged N-terminal fragment of ICE1 (NL: 1-1190). Source data are provided as a Source Data file. MW (kDa) indicates molecular weight.

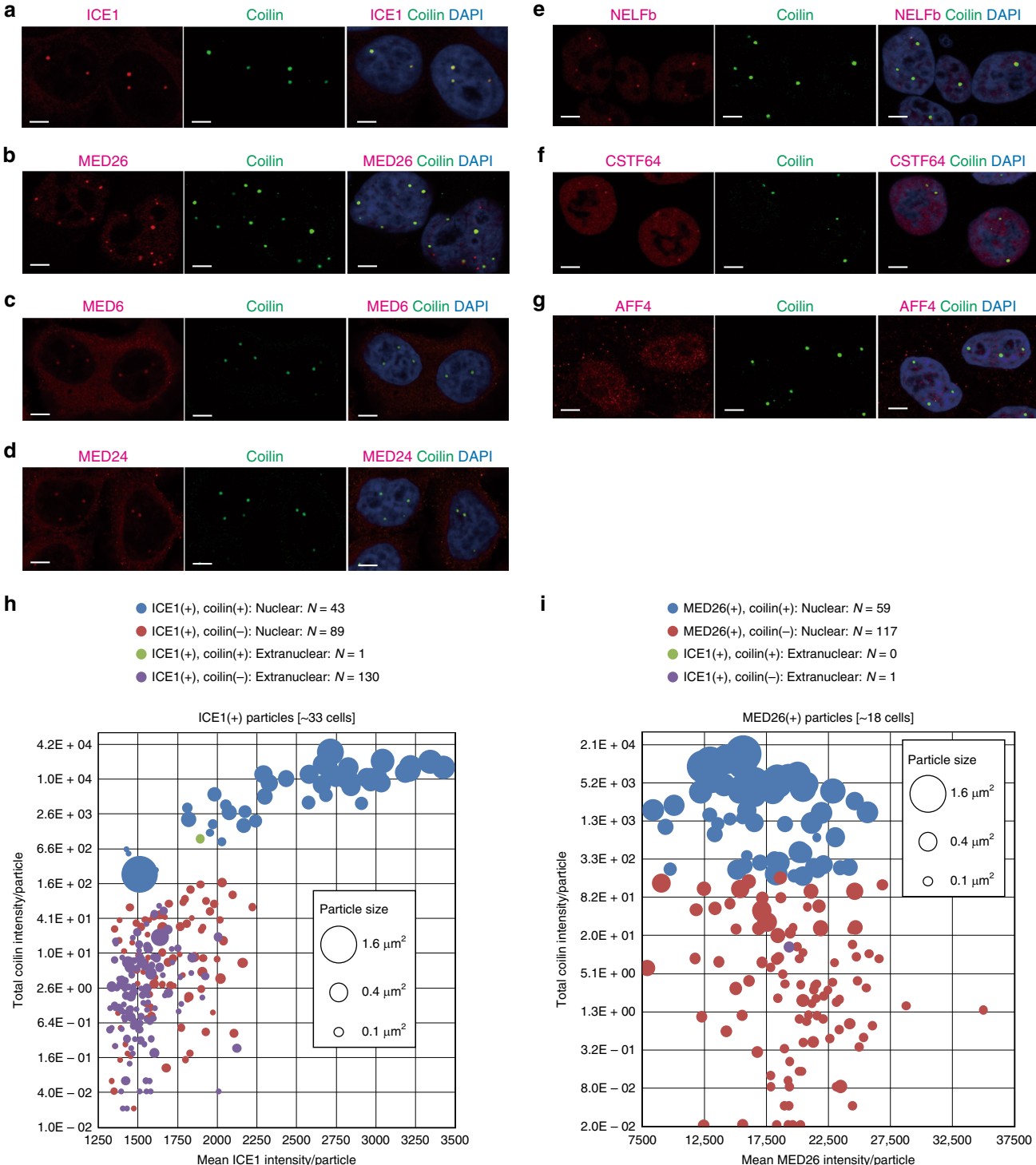

**Fig. 7 Colocalization of the components of Mediator, LEC, NELF and HLF in Cajal bodies. a–g** HeLa cells were fixed by formaldehyde and stained with anti-ICE1 (magenta), anti-MED26 (magenta), anti-MED6 (magenta), anti-MED24 (magenta), anti-NELFb (magenta), anti-CSTF64 (magenta), anti-AFF4 (magenta), and anti-coilin (green) antibodies. Scale bars, 5 μm. **h** Quantification of the number and signal intensity of ICE1 particles in HeLa cells. **i** Quantification of the number and signal intensity of MED26 particles in HeLa cells.

occupancy of INTS9 was decreased by knockdown of ICE1 (Fig. 9b). Consistent with our notion that LEC contributes to recruitment of 3′-end processing factors, knockdown of ICE1 did not affect the protein levels of CPSF3, CSTF50, and INTS9 (Supplementary Fig. 8a). Next, we examined whether knockdown of ICE1 affected the occupancy of Pol II at the genes. As shown in Supplementary Fig. 8b, ICE1 knockdown did not affect the

occupancy of Pol II at RDH genes, indicating that the decrease in CPSF3 recruitment by knockdown of ICE1 is not simply an indirect effect of a decrease in Pol II at RDH genes. In contrast, ICE1 knockdown moderately decreased the occupancy of Pol II at snRNA genes (Supplementary Fig. 8c); however, Pol II occupancy was reduced to a lesser degree than that of CPSF3 (Fig. 9b). Hence, our results argue that the decrease in INTS9 recruitment

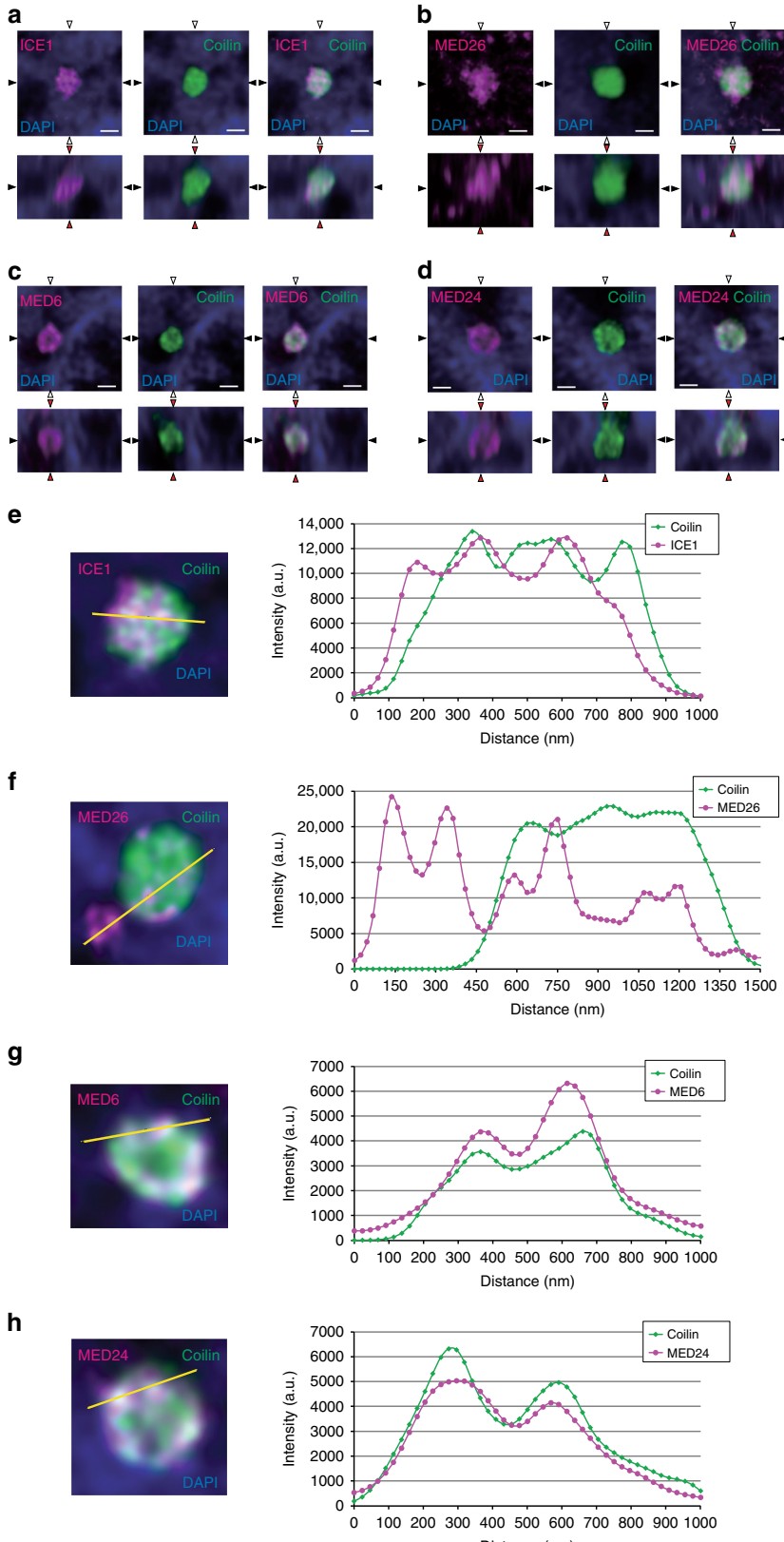

**Fig. 8 Super-resolution imaging of Mediator and LEC in Cajal bodies. a–d** HeLa cells stained with anti-ICE1 (magenta), anti-MED26 (magenta), anti-MED6 (magenta), and anti-coilin (green) antibodies were observed by stimulated emission depletion (STED) microscopy. Scale bars, 0.5 μm. X axis, Y axis, and Z axis are represented by black triangle, white triangle, and red triangle, respectively. **e–h** Line intensity plots across Cajal bodies (shown in **a–d**) of HeLa cells stained with anti-ICE1 (magenta), anti-MED26 (magenta), anti-MED6 (magenta), and anti-coilin (green) antibodies.

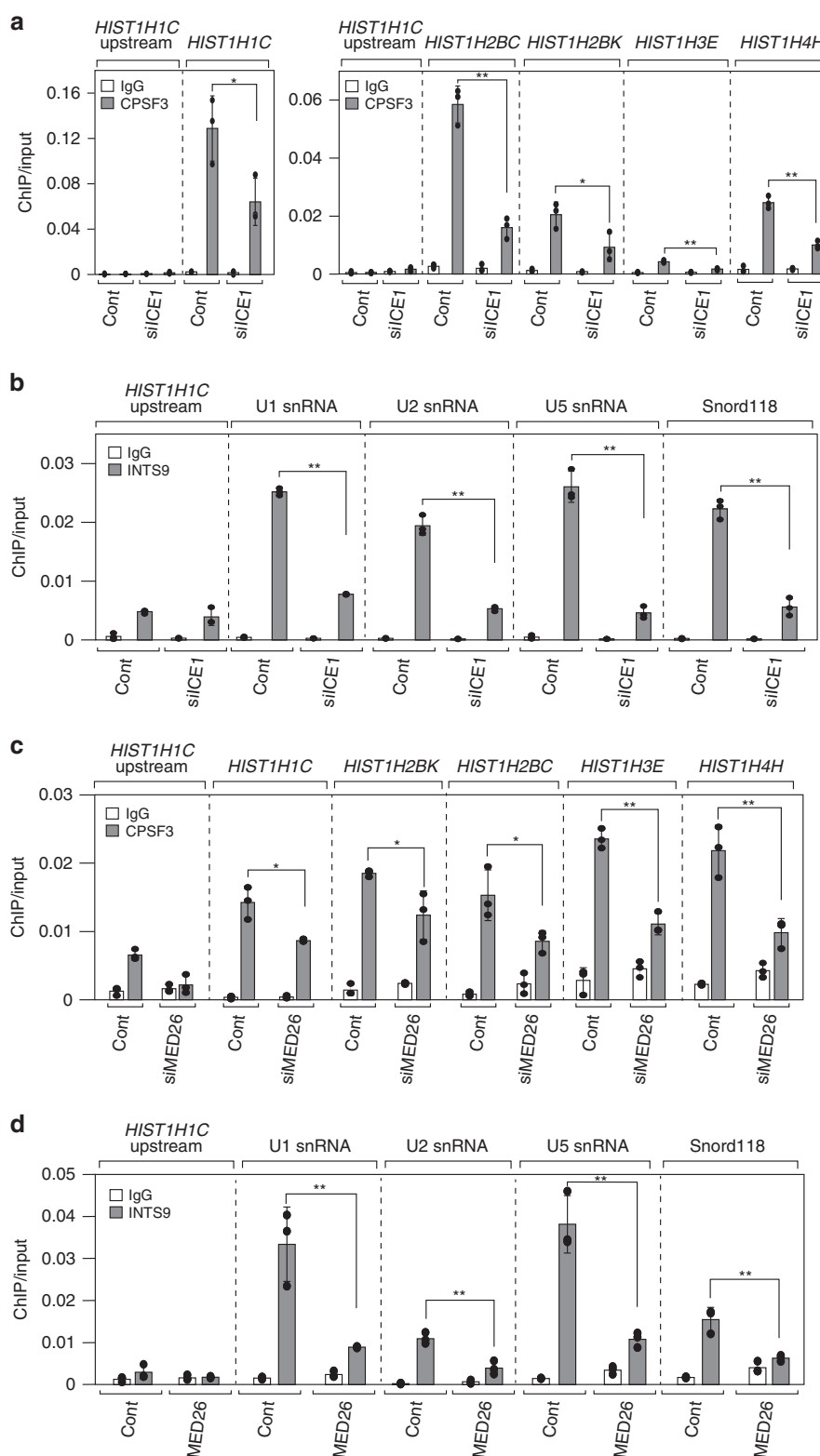

to snRNA genes cannot be attributed simply to an indirect effect of a decrease in Pol II, but also reflects a direct effect of the decrease in ICE1 at snRNA genes. Consistent with these results, ICE1 did not affect steady-state levels of total RDH transcripts, but moderately decreased the total transcripts of the U1 snRNA gene (Supplementary Fig. 8d). In addition, we found that knockdown of MED26 decreased CPSF3 and INTS9 occupancy at RDH and snRNA genes, respectively (Fig. 9c, d). These results

suggest that knockdown of MED26 interferes with multiple processes, including transcription initiation, elongation, and termination and transcript 3′-processing, while knockdown of ICE1 is likely to interfere with transcription termination and 3′ processing only, consistent with the result of the polyA-selected RNA-seq showing that the transcription termination defect by MED26 knockdown was greater than that by ICE1 knockdown (Fig. 2a, e). Thus, these results show that LEC brought by

**Fig. 9 ICE1 is required for occupancy of the components of HLF and Integrator. a** Knockdown of ICE1 decreased the occupancy of CPSF3 at RDH genes. Ct values of each ChIP were normalized to that of input. Each value is the average of three independent experiments and error bars show standard deviation. The P values for the indicated comparisons were determined by Student's t test (*P < 0.05; **P < 0.01). n = 3 biologically independent samples. Source data are provided as a Source Data file. **b** Knockdown of ICE1 decreased the occupancy of INTS9 at snRNA genes. Ct values of each ChIP were normalized to that of the input. Each value is the average of three independent experiments and error bars show the standard deviation. The P values for the indicated comparisons were determined by Student's t test (*P < 0.05; **P < 0.01). n = 3 biologically independent samples. Source data are provided as a Source Data file. **c** Knockdown of MED26 decreased the occupancy of CPSF3 at RDH genes. The Ct values of each ChIP were normalized by that of the input. Each value is the mean of three independent experiments and error bars show the standard deviation. The P values for the indicated comparisons were determined by Student's t test (*P < 0.1; **P < 0.05). n = 3 biologically independent samples. Source data are provided as a Source Data file. **d** Knockdown of MED26 decreased the occupancy of INTS9 at snRNA genes. The Ct values of each ChIP were normalized by that of the input. Each value is the mean of three independent experiments and error bars show the standard deviation. The P values for the indicated comparisons were determined by Student's t test (*P < 0.05; **P < 0.01). n = 3 biologically independent samples. Source data are provided as a Source Data file.

MED26 supports recruitment of HLF or Integrator to RDH or snRNA genes, respectively.

Because the snRNA promoter was shown to be required for proper transcription termination and 3′-processing of snRNA genes in *Drosophila*[40–42]. In addition, our results are consistent with the model that LEC recruited to the vicinity of RDH gene promoters by MED26-containing Mediator helps to regulate termination and 3′-processing of RDH transcripts, raising the possibility that these processes might be influenced by promoter sequences. To address this possibility, we examined how replacement of the HIST1H1C promoter with the cytomegalovirus (CMV) promoter, a well-characterized viral promoter that normally directs transcription of a polyadenylated transcript, affects 3′-end processing of an RDH transcript. For this, we performed a plasmid-based assay to determine the effects of promoter exchange in cells. As shown in Supplementary Fig. 9a, we generated two plasmids that contain either the HIST1H1C promoter or CMV promoter, the *HIST1H1C* gene region from TSS to 100 bp downstream of TES, including the *HIST1H1C* stem loop and PAS. In addition, we added a FLAG-tag sequence at the C-terminus of the *HIST1H1C* coding region to distinguish plasmid-encoded histone H1 from endogenous protein. We transiently transfected the plasmids and compared the levels of total transcript, unprocessed read-through transcript, and protein when transcription is driven by the *HISTH1C* or the CMV promoter. As shown in Supplementary Fig. 9b, c, replacing the HIST1H1C promoter with the CMV promoter increased the level of unprocessed read-through transcripts, but not total transcripts, suggesting that events occurring at the promoter region contribute to the decision to utilize the stem-loop-dependent or PAS-dependent 3′ processing and termination pathways. This result is consistent with our notion that LEC recruited by MED26-containing Mediator to RDH genes interferes with read-through of Pol II at these genes. Intriguingly, replacement of the HIST1H1C promoter with the CMV promoter also increased the protein levels of HIST1H1C-FLAG (Supplementary Fig. 9d), consistent with previous studies showing that polyadenylation of histone mRNAs increases their stabilities and can lead to increased levels of histone proteins[43].

Our results provided hints toward answering the important question of the overall biological significance of Mediator and LEC regulation in 3′-end processing of RDH genes. Non-polyadenylated histone mRNAs were shown to be regulated to be stable only during the S phase to prevent harmful production of free histone proteins in cells outside S phase, because excess histone levels can lead to cytotoxicity through multiple mechanisms[7,44–47]. Degradation of SLBP by the ubiquitin-proteasome pathway outside S phase is associated with disappearance of histone mRNAs outside the S phase[7,48]. Considering that SLBP also copurified with ICE1 (Fig. 6a) and has been shown to associate with CBCA-NELF-DSIF[19], our results suggest that LEC

and SLBP could play a role in interfering with polyadenylation of RDH mRNAs to prevent inappropriate production of free histone proteins in cells outside the S phase.

Based on our results, we propose a model that (i) MED26 recruits LEC to snRNA and RDH genes and plays a role in transcription processes, including initiation and elongation, then (ii) LEC binds to CBCA-NELF-DSIF and cooperatively inhibits reading through by Pol II at the genes, and finally (iii) LEC promotes 3′-end processing of the RDH genes and snRNA genes through the recruitment of 3′-end processing factors for RDH genes or Integrator complex, respectively (Fig. 10).

## Discussion

In this report, we present evidence that MED26-containing Mediator recruits LEC through interaction with the NTD of MED26 and regulates the transcription termination and 3′-end processing at a subset of genes that produce non-polyadenylated snRNA and RDH transcripts. We show evidence that both MED26 and LEC are present at snRNA and RDH genes. Intriguingly, we observed that transcription termination at snRNA and RDH genes is defective in cells expressing MED26 lacking its NTD, which we have shown previously is required for recruitment of LEC to snRNA genes[24]. Proteomic analysis revealed that LEC copurifies with CBCA, NELF and DSIF. Several reports have demonstrated that CBCA and NELF/DSIF play roles in terminating the transcription of snRNA and RDH genes[14,15,19]. In addition, we found that LEC copurified with HLF and Integrator, which are 3′-processing factors for the snRNA and RDH genes, respectively. Consistent with these results, we observed that components of Mediator, LEC, NELF, and HLF are colocalized at Cajal bodies, which are involved in transcription regulation of snRNA and RDH genes[7,19,31,34,38,49]. Super-resolution imaging revealed that Mediator and LEC formed similar structures in Cajal bodies, consistent with our proposal that Mediator and LEC play a role in transcription termination of non-polyadenylated genes at Cajal bodies. Of note, knockdown of ICE1 in cells decreased the recruitment of HLF to RDH genes and Integrator to snRNA genes. Based on our findings, we propose a model in which MED26 and LEC regulate termination of snRNA genes and RDH transcripts. In this model, LEC is recruited by MED26 to the gene. At the termination step of transcription, LEC binds to CBCA, NELF, and DSIF and inhibits the reading through of Pol II at the genes. Then, LEC recruits HLF or Integrator to the RDH or snRNA genes, respectively, to promote their 3′-end processing. Our results indicate that MED26 plays a role in the recruitment of LEC to specifically regulate the genes encoding non-polyadenylated transcripts. Considering that MED26 is a metazoan-specific subunit of the Mediator complex and conserved from *Drosophila melanogaster* to humans, but is not present in yeast, we expect that the mechanism uncovered in our study machinery is metazoan-specific.

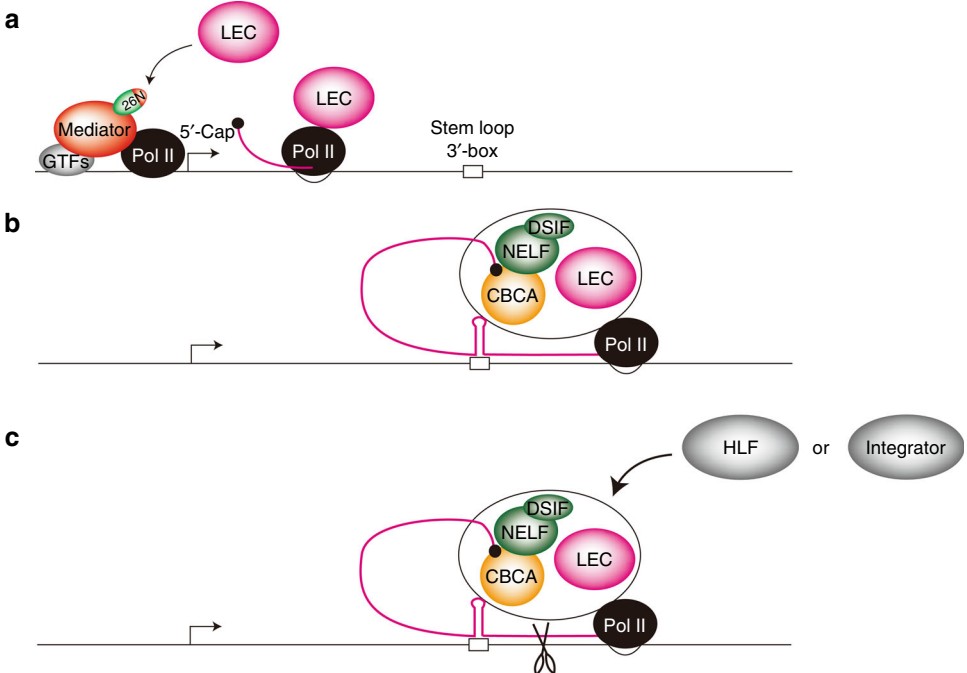

**Fig. 10 Model for the role of MED26 and LEC in transcription termination. a** MED26 recruits LEC to snRNA and RDH genes. **b** LEC binds to CBCA-NELF-DSIF and cooperatively inhibits reading through by Pol II at the genes. **c** LEC promotes 3′-end processing of the RDH and snRNA genes through the recruitment of HLF or Integrator, respectively.

Prior studies have provided evidence that yeast Mediator subunit MED18 contributes to recruitment of the cleavage and polyadenylation factors Rna15 and Pta[50] and proper transcription termination at several genes. Our results showing that MED26 also promotes transcription termination and 3′ processing at snRNA and RDH genes, through the recruitment of LEC, provides further support for the idea that Mediator can modulate transcriptional events distal from the promoter.

This study and our previous research raise the possibility that NTD of MED26 plays a role in the recruitment of SEC to a subset of genes encoding mRNAs containing poly(A) tails and LEC to a subset of genes encoding mRNAs lacking poly(A). However, this hypothesis does not completely explain the mechanisms by which target genes of SEC and LEC are determined because EAF directly interacts with NTD of MED26 and is a shared component of both SEC and LEC[23,24]. It is conceivable that other components of SEC or LEC also contribute to determination of the target genes of SEC or LEC. A recent study showed that ENL, a component of SEC, functions as a chromatin reader and binds to H3K27ac and H3K9ac of the promoter via YEATS domain of ENL[51]. It is thought that MED26 and ENL cooperatively contribute to SEC recruitment to the target genes. It is also possible that an inter-actor or component of LEC functions as a chromatin reader and helps MED26 recruit LEC to its targets, including snRNA and RDH genes.

Our results indicate that LEC recruits HLF or Integrator and promotes the 3′-processing of RDH or snRNA transcript, respectively. However, it remains unclear how LEC recruits two distinct complexes to the different genes. In RDH genes, the stem-loop region located at the 3′ end of pre-mature mRNA is bound by SLBP and HDE is recognized by U7 snRNP[6,7]. Considering that SLBP and U7 snRNP can help to recruit HLF to genes[6,8], it is speculated that LEC, SLBP, and U7 snRNP cooperatively recruit HLF to RDH genes. It has been shown that histone mRNAs are only stable during the S phase, because SLBP, which is only present during the S phase, binds to the stem-loop structure of RDH transcripts to prevent RDH mRNAs from undergoing degradation

during the S phase[7]. Degradation of SLBP by the ubiquitin-proteasome system outside the S phase is associated with dis-appearance of histone mRNAs outside the S phase[47,48]. Further-more, our plasmid-based assay revealed that replacement of a non-polyadenylated gene promoter (HIST1H1C promoter) with a polyadenylated gene promoter (CMV promoter) resulted in increased levels of read-through RDH transcripts and proteins. Considering that SLBP copurified with ICE1 (Fig. 6a) and was shown to be a component of CBCA-NELF-DSIF[19], our results raise the possibility that MED26-containing Mediator and LEC play a role in interfering with polyadenylation of RDH mRNAs to prevent harmful production of free histone proteins in cells outside the S phase[43,45]. In the case of snRNA genes, it has been shown that the 3′-box element located at the 3′ end of pre-mature snRNAs is bound by Integrator[5,13], indicating that LEC and the 3′-box ele-ment help to recruit Integrator to snRNA genes. In addition, substitution of the snRNA promoter with the actin promoter was shown to interfere with proper 3′-end formation of snRNA genes in *Drosophila*, indicating that the promoter is also required for proper transcription termination of snRNA genes[40].

Based on our findings and previous reports, we propose a model in which (i) MED26 recruits LEC to snRNA and RDH genes and plays a role in transcription processes, including initiation and elongation, (ii) LEC binds to CBCA and NELF/DSIF to increase Pol II termination, and (iii) LEC promotes the 3′ processing of RDH or snRNA genes through the recruitment of HLF or Integrator, respectively.

## Methods

**Cell culture and cell lines.** Human embryonic kidney 293T (HEK293T) cells and derivatives, HeLa S3 cells and their derivatives, and Flp-In 293 cells (Invitrogen, Carlsbad, CA) and their derivatives were cultured under an atmosphere of 5% $CO_2$ at 37 °C in Dulbecco's modified Eagle's medium (Sigma-Aldrich Corp., St. Louis, MO) supplemented with 10% (v/v) fetal bovine serum (Invitrogen), 55 μM β-mercaptoethanol (Gibco, Grand Island, NY), 2 mM L-glutamine, penicillin (10 U ml$^{-1}$), and streptomycin (0.1 mg ml$^{-1}$). No mycoplasma contamination in these cell lines was confirmed using a PCR Detection Kit (EZ-PCR™ Mycoplasma Detection Kit, Cat# 20-700-20; Biological Industries, CT, USA). Flp-In 293 cells

stably expressing FLAG-tagged ICE1 and AF9 were generated previously[24]. The MED26 hypomorphic HEK293T cell line D2G8 was generated by applying two small guide RNAs (sgRNAs) complementary to regions of exon 3 in the MED26 gene to induce random sequence insertions and/or deletions. Both strands (sense and antisense) of sgRNA sequences targeting MED26 were chemically synthesized, annealed, phosphorylated using T4 polynucleotide kinase (NEB, #0201S), and ligated into the *Bbs*I site downstream of the U6 promoter in pX458/459 plasmids (Addgene, #48138/48139). Oligonucleotide sequences corresponding to the guide RNAs are included in the list of oligonucleotides in Supplementary Information. CRISPR-mediated mutagenesis was performed essentially as described[52]. The resulting MED26 hypomorphic clonal cell line, D2G8, contains an allele with a 140 bp deletion that removes sequences encoding MED26 NTD helix 4, which is critical for interaction with SEC and LEC components EAF1 or EAF2[23,53]. It also contains alleles with frameshift deletions in exon 3 that prevent expression of full-length MED26 but that contain in-frame CUG codons, which can support cryptic CUG/leucine initiation[54], upstream of the most N-terminal MED26 peptide identified in Mediator enriched from these cells (Supplementary Fig. 4). HCT116 cells (ATCC, CCL-247) and HCT116-MED26-AID were cultured in McCoy's 5A medium supplemented with 10% fetal bovine serum, penicillin–streptomycin (100 U ml⁻¹ each), and Glutamax. HCT116-MED26-AID cells were additionally supplemented with puromycin (10 µg ml⁻¹) and hygromycin (100 µg ml⁻¹). The HCT116-MED26-AID cell line was established as described[55]. Briefly, HCT116 cells expressing auxin-dependent F-box protein TIR1 (HCT116-OsTIR1 cells) were generated by introducing OsTIR1 into the AAVS1 safe harbor locus in HCT116 cells by co-transfecting pMK232 (CMV-OsTIR1-PURO) and AAVS1 T2 CRISPR in pX330. After isolation of puromycin-resistant clones, OsTIR1 expression was confirmed by immunoblotting. The AID-hygromycin cassette in pMK287 (mAID-Hygro) was modified by inserting a DNA fragment encoding a linker sequence (EPTTEDLYFQSDNSSPGSGAGA) at its 5′ end to create linker-AID-hygromycin cassette. Standard molecular biology methods were used to generate a donor fragment for CRISPR-mediated homologous recombinational insertion of the linker-AID-hygromycin cassette at the 3′ end of the MED26 coding sequence in chromosome 19. HCT116-OsTIR1 cells were co-transfected with the donor fragment and pX459 derivative encoding Med26 gRNA (GUCGAUCCGCACGGCGA CGAC). After hygromycin clonal selection, correct insertion of the linker-AID-hygromycin cassette into both Med26 alleles was confirmed by PCR and sequencing. pMX232 (Addgene, #72834), AAVS1 T2 CRISPR in pX330 (Addgene, #72833), and pMX287 (Addgene, #72825) were gifts from Masato Kanemaki, and pX459 (Addgene #62988) was a gift from Feng Zhang. To synchronize the cell cycle at the S phase, we cultured wild-type and MED26 hypomorphic HEK293T cells in medium containing 3 µg ml⁻¹ APH (A0781; Sigma-Aldrich Corp.) for 24 h. The cells were subsequently released from the S phase by exchanging the medium and harvested at specified times.

**siRNA transfection**. HEK293T cells in six-well tissue culture plates (~1 × 10⁵ cells per well) or 10-cm dishes (~2 × 10⁶ cells per dish) were transfected with 50 nM siRNAs targeting human MED26 (#1, s18074; #2, s18075; #3, s18076; Ambion/Life Technologies, ON-TARGET plus SMART pool, L-011948-00, Dharmacon, Pittsburgh, PA), siRNA targeting human ICE1 (ON-TARGET plus SMART pool, L-024272-02; Dharmacon), siRNA targeting human AFF4 (ON-TARGET plus SMART pool, L-020276-00; Dharmacon), siRNA targeting human CDK9 (ON-TARGET plus SMART pool, L-003243-00; Dharmacon), or with 50 nM siGEN-OME NON-TARGETING siRNA Pool #2 (D-001206-14; Dharmacon) using Lipofectamine™ RNAiMAX Transfection Reagent (Invitrogen).

**Production of recombinant proteins**. N-terminally 6× His- and FLAG-tagged SLBP, CBP80, and CBP20 and N terminally 6× His- and HA-tagged ICE1-NL (1-1190) were subcloned into pFastBac HTb (Life Technologies, Carlsbad, CA) and expressed singly or together with the BAC-to-BAC system (Clontech).

**Western blotting**. Anti-FLAG M2 antibodies (1:2000 dilution; Sigma-Aldrich Corp.), anti-HA antibodies (1:2000 dilution; Covance, Princeton, NJ), anti-MED26 antibodies (1:1000 dilution, sc-48776 X; Santa Cruz Biotechnology, Santa Cruz, CA), anti-MED26 antibodies (1:1000 dilution, 14950S; Cell Signaling), anti-MED1 antibodies (1:1000 dilution, sc-5334 X; Santa Cruz), anti-MED23 antibodies (1:1000 dilution, A300-425A; Bethyl Laboratories), anti-ICE1 antibodies (1:200 dilution, HPA054452; Sigma-Aldrich Corp.), anti-CBP80 antibodies (1:1000 dilution, 24965S; Cell Signaling Technology, Beverly, MA), anti-CBP20 antibodies (1:200 dilution, sc-48793; Santa Cruz), anti-ZC3H18 antibodies (1:1000 dilution, A304-682A; Bethyl Laboratories, Montgomery, TX), anti-ARS2 antibodies (1:1000 dilution, ab192999; Abcam, Cambridge, UK), anti-SLBP antibodies (1:1000 dilution, RN045P; Medical & Biological Laboratories Co., Ltd., Nagoya, Japan), anti-NELFb antibodies (COBRA1, D6K9A, #14894S; Cell Signaling Technology), anti-NELFe antibodies (1:200 dilution, ab170104; Abcam), anti-Symplekin antibodies (1:1000 dilution, ab80274; Abcam), anti-CPSF1 antibodies (1:200 dilution, ab81552; Abcam), anti-CSTF50 antibodies (1:1000 dilution, A301-250A; Bethyl Laboratories), anti-CSTF64 antibodies (1:1000 dilution, A301-092A; Bethyl Laboratories), anti-CPSF3 (CPSF73) antibodies (1:200 dilution, A301-091A; Bethyl Laboratories), anti-INTS4 antibodies (1:1000 dilution, ab75253; Abcam),

anti-INTS9 antibodies (1:1000 dilution, 13945S; Cell Signaling Technology), anti-ZC3H8 antibodies (1:1000 dilution, ab97821; Abcam), anti-CDK9 antibodies (1:200 dilution, sc-13130; Santa Cruz), anti-nucleolin antibodies (1:1000 dilution, ab13541; Abcam), and anti-AFF4 antibodies (1:1000 dilution, A302-538A; Bethyl Laboratories) were used in western blots. As secondary antibodies, anti-mouse IgG antibodies (1:2000 dilution, A4416; Sigma), anti-rabbit IgG antibodies (1:3000 dilution, NA9340V; GE Healthcare), anti-goat IgG antibodies (1:3000 dilution, A4174; Sigma), anti-rabbit IgG antibodies (Dylight 800, 611-145-122; Rockland Immunochemicals Inc., PA), and anti-mouse IgG antibodies (Dylight 800, 610-145-121; Rockland Immunochemicals Inc.) were used.

**Immunoprecipitation and affinity purification**. Protein complexes were purified from nuclear extract fractions of cell lines stably expressing FLAG-tagged proteins using anti-FLAG M2 agarose (E2220, Sigma-Aldrich Corp.)[24]. Briefly, nuclear extracts and S100 fractions were prepared in the presence of Benzonase® Nuclease (E8263; Sigma-Aldrich Corp.), basically in accordance with the method of Dignam et al.[56] from parental HeLa cells or HEK293 FRT cells stably expressing FLAG-tagged proteins. Each of the nuclear extracts was incubated with 100 µl of anti-FLAG agarose beads for 2 h at 4 °C. The beads were washed five times with a 100-fold excess of a buffer containing 50 mM HEPES-NaOH (pH 7.9), 0.15 M NaCl, 0.1% Triton X-100, and 10% (v/v) glycerol and then eluted with 100 µl of a buffer containing 0.1 M NaCl, 50 mM HEPES-NaOH (pH 7.9), 0.05% Triton X-100, 10% (v/v) glycerol, and 0.25 mg ml⁻¹ FLAG peptide. Endogenous ICE1-interacting proteins were affinity-purified using anti-ICE1 antibodies #1 (HPA054452; Sigma-Aldrich Corp.) or #2 (A304-276A; Bethyl Laboratories). Nuclear extracts of HEK293T cells were incubated with 10 µg of normal rabbit IgG (PM035; Medical and Biological Laboratories, Nagoya, Japan) or each of the ICE1 antibodies bound to protein A magnetic beads (161-4011; Bio-Rad, Hercules, CA) for 2 h at 4 °C. After washing the beads with a buffer (40 mM HEPES-NaOH, pH 7.9, 0.15 M NaCl, 0.1% Triton X-100), the immunoprecipitates were eluted from the beads by addition of sodium dodecyl sulfate (SDS) sample buffer and subjected to SDS-–polyacrylamide gel electrophoresis (SDS-PAGE) and western blot analysis.

**MudPIT analysis**. FLAG immunoaffinity- (Fig. 5c) or glutathione affinity- (Supplementary Fig. 4) purified proteins and negative controls were treated with benzonase and trichloroacetic acid precipitated before analysis by MudPIT[57,58]. After denaturation, reduction, and alkylation, proteins were digested with endoproteinase LysC followed by trypsin (Promega). Peptides mixtures were analyzed through 10 multidimensional liquid chromatography steps implemented on a quaternary Agilent 1100 series HPLC in line with a Thermo linear ion trap mass spectrometer. Tandem mass (MS/MS) spectra acquired for the analyses of proteins purified by glutathione affinity (Supplementary Fig. 4) were interpreted using SEQUEST (v. 27)[59,60] against a database of non-redundant human proteins downloaded from NCBI on 3-25-2015. The MS/MS dataset acquired for the analyses of proteins purified by FLAG immunoaffinity (Fig. 5c) were interpreted using ProLuCID (v. 1.3.3)[60], against a database of NR proteins downloaded from NCBI on 6-10-2016. Both databases were complemented with sequences from usual contaminants (human keratins, IgGs, proteolytic enzymes). To estimate FDRs, each sequence was randomized keeping amino acid composition and length the same, and the resulting "shuffled" sequences were added to the forward sequences and searched at the same time. Mass tolerance for both precursor and fragment ions were set at 3 a.m.u. and 800 p.p.m. in the SEQUEST and ProLuCID searches, respectively, and to account for alkylation by CAM, 57 Da were added statically to the cysteines. The SEQUEST searches were performed without any peptide end requirements and differential modifications, while the ProLuCID searches were set up against a preprocessed database of tryptic peptides with K/R at both ends and with a differential modification of +16 Da on methionine residues. No maximum number of missed cleavages were specified. Peptide/spectrum matches were sorted and selected using DTASelect/CONTRAST (v. 1.9)[61] in combination with an in-house software, *swallow* (v. 0.0.1, https://github.com/tzw-wen/kite), to filter spectra, peptides, and proteins at FDRs <0.5%. Combining all runs, proteins had to be detected by at least two such peptides or by one peptide with two independent spectra. To estimate relative protein levels, distributed normalized spectral abundance factors (dNSAFs) were calculated[57,62–64]. The dNSAF for a protein $k$ is proportional to the amount of the protein present in the sample and is calculated by the formula:

$$\mathrm{dNSAF}_i = \frac{\mathrm{dSAF}_i}{\sum_{i=1}^{N} \mathrm{dSAF}_i}, \quad (1)$$

with

$$\mathrm{dSAF}_i = \frac{\mathrm{uSpC}_i + \frac{\mathrm{uSpC}_i}{\sum_{m=1}^{M} \mathrm{uSpC}_m} \times \mathrm{sSpC}_{it}}{\mathrm{Length}_i}, \quad (2)$$

in which shared spectral counts (sSpCs) are distributed based on spectral counts unique to each protein $i$ (uSpC) divided by the sum of all unique spectral counts for the M protein isoforms that shared peptide $j$ with protein $i$.

**ChIP assays**. Cells from one 10-cm dish (~1 × 10⁷) of HEK293T cells grown to 80% confluence were used for immunoprecipitation. The cells were cross-linked

with 2 mM DSG Crosslinker (c1104; ProteoChem, Loves Park, IL) in phosphate-buffered saline (PBS) for 30 min and then 1% formaldehyde in PBS for 20 min at room temperature. Then, the cells were resuspended and lysed in lysis buffer (0.2% or 0.5% SDS, 10 mM EDTA, 150 mM NaCl, 50 mM Tris-HCl, pH 8.0), and were then sonicated with a Bioruptor® Sonicator (Diagenode, Denville, NJ) 20 times for 30 s each at the maximum power setting to generate DNA fragments of ~150–500 bp. For Rpb1 ChIP, cells were cross-linked with 1% formaldehyde in PBS for 20 min at room temperature before quenching with glycine to a final concentration of 0.125 M, washed twice with ice-cold PBS, and resuspended in lysis buffer (15 mM HEPES, pH 7.5, 140 mM NaCl, 1 mM EDTA, 0.5 mM EGTA, 1% Triton X-100, 0.1% NaDOC, 1% SDS, 0.5% N-lauroylsarcosine, 1 mM dithiothreitol (DTT) and 1:100 Protease inhibitor cocktail (Sigma). Mouse L cells (ATCC CRL-2648, 10% of human cell number) were added to the human cells as a spike-in control before sonication. Combined cells were sonicated using a Misonix 3000 sonicator at 4 °C using output 2.5 (9 w power) for 10 cycles (10 s ON/60 s OFF) to generate DNA fragments of ~150–500 bp. Sonicated chromatin was incubated at 4 °C overnight with 5–10 µg of normal IgG or specific antibodies. The specific antibodies used were as follows: MED26 (H-228, sc-48776 X; Santa Cruz), ICE1 (A304-276A; Bethyl Laboratories), ELL (A301-645A; Bethyl Laboratories), CPSF3 (A301-091A; Bethyl Laboratories), INTS9 (anti-RC74, A300-412A; Bethyl Laboratories), and Rpb1 (D8L4Y, 14958S; Cell Signaling Technology). Then, Dynabeads™ Protein A for Immunoprecipitation (10001D; ThermoFisher Scientific) was added and incubated for 2 h at 4 °C. The beads were washed two times with IP buffer (20 mM Tris-HCl, pH 8.0, 150 mM NaCl, 2 mM EDTA, 1% Triton X-100), two times with high-salt buffer (20 mM Tris-HCl, pH 8.0, 500 mM NaCl, 2 mM EDTA, 1% Triton X-100), once with LiCl buffer (250 mM LiCl, 20 mM Tris-HCl, pH 8.0, 1 mM EDTA, 1% Triton X-100, 0.1% NP40, and 0.5% NaDOC), and two times with TE buffer. Bound complexes were eluted from the beads with 100 mM NaHCO₃ and 1% SDS by incubation at 50 °C for 30 min with occasional vortexing. Crosslinking was reversed by overnight incubation at 65 °C. Immunoprecipitated DNA and input DNA were treated with RNase A and proteinase K by incubation at 45 °C. DNA was purified using the QIAquick PCR Purification Kit (28106; Qiagen) or MinElute PCR Purification Kit (28006; Qiagen). Immunoprecipitated and input material was analyzed by qPCR. The ChIP signal was normalized to the total input. Three biological replicates were performed for each experiment. Primer sets are detailed in the Supplementary Information.

**ChIP-seq and gene annotation.** Sequencing reads were acquired through primary Solexa image analysis. Filtered reads were then aligned either to the human genome (hg38) for MED26 ChIP or to an mm10-hg38 combined genome for Rpb1 ChIP using the Bowtie alignment tool. Only those sequences that matched uniquely to the genome with up to two mismatches and mapped to fewer than three locations were retained for subsequent analyses. Sequence reads for each ChIP-sequence dataset and its associated whole-cell extract controls were used for input. ChIP-seq genome browser tracks depict the average sequence reads from duplicate ChIPs. The public data were downloaded from GEO database with accession ID GSE47938. Reads were aligned to human genome hg38 using Bowtie2 (version 2.2.4) with default settings. Peaks were called using MACS2 callpeak (version 2.1.1), default parameter, based on the corresponding input samples. Peaks with q-score <1e − 4 were used for downstream analysis.

**RNA-seq analysis and RT-qPCR.** Total RNA was isolated using miRNeasy Mini Kit (217004; Qiagen). For RT-qPCR, total mRNA was reverse transcribed using the iScript Select cDNA Synthesis Kit (1708897; Bio-Rad). The threshold cycle (Ct) values were determined by real-time PCR reactions using an Applied Biosystems StepOne Real-time PCR System and Power SYBR Green PCR Master Mix (Life Technologies) and normalized by subtracting the Ct value of the CT from the Ct value of the UT ($\Delta Ct = Ct^{UT} - Ct^{CT}$). The relative unprocessed and total transcript levels were then calculated using $2^{-\Delta Ct}$. Primer sequences are listed in the Supplementary Information. For oligo-dT selection-based RNA-seq analysis or ribo-depleted RNA-seq analysis, 1 µg of total RNA was subjected to oligo-dT selection or depletion of ribosomal RNA using the Ribo-Zero Kit (MRZH11124; Illumina, CA), and libraries were prepared using the TruSeq RNA Sample Prep Kit (Illumina). Raw reads from sequencing were demultiplexed allowing up to one mismatch using Illumina bcl2fastq2 v2.18. Reads were then mapped to human genome hg38 with STAR aligner (version 2.5.3a) default settings, using Ensembl 87 gene annotation models. Transcripts per kilobase million values were then generated using RSEM (version 1.3.0) function rsem-calculate-expression with option –estimate-rspd. Differential gene expression analysis was performed using R (v. 3.22.3) package edgeR (v. 3.5.0). Genes with FDR <0.05 and absolute fold change >1.5 were included for downstream analysis.

**PRO-seq analysis.** Nuclei were isolated from HEK293T cells and mutant cell lines[65,66]. Briefly, 25 million cells were collected and washed with ice-cold PBS. To reduce sample processing bias, D. melanogaster Kc167 cells (10% of human cell number) were added to each sample as a spike-in control. The combined cells were resuspended in cold douncing buffer (10 mM Tris-HCl, pH 7.4, 300 mM sucrose, 3 mM CaCl₂, 2 mM MgCl₂, 0.1% Triton X-100, 0.5 mM DTT, 1:100 Protease inhibitor cocktail (Sigma) and 4 U ml⁻¹ RNase inhibitor; SUPERaseIN) and dounced

30 times until 90% of cells were lysed. Isolated nuclei were pelleted, washed with douncing buffer, and resuspended in ice-cold storage buffer (10 mM Tris-HCl, pH 8.0, 25% glycerol, 5 mM MgCl₂, 0.1 mM EDTA, and 5 mM DTT) to 2 × 10⁷ nuclei per 100 µl. Nuclear run-on (NRO) assays were performed with biotin-11-NTPs. In all, 2 × 10⁷ nuclei per 100 µl were thoroughly mixed with equal amount of pre-heated 2 × NRO reaction mixture (10 mM Tris-HCl, pH 8.0, 5 mM MgCl₂, 300 mM KCl, 1 mM DTT, 1% Sarkosyl, 50 µM each of Biotin-11-A/G/C/UTP (PerkinElmer), 0.8 U µl⁻¹ RNase inhibitor) and incubated at 37 °C for 3 min in a heat block. Nascent RNA was extracted, purified, and fragmented by base hydrolysis in 0.2 N NaOH on ice for 10 min. After neutralization, fragmented nascent RNA was bound to Dynabeads™ M-280 Streptavidin magnetic beads (ThermoFisher Scientific) and incubated for 20 min at 4 °C. The beads were sequentially washed twice in high salt (2 M NaCl, 50 mM Tris-HCl, pH 7.4, 0.5% Triton X-100), twice in medium salt (300 mM NaCl, 10 mM Tris-HCl, pH 7.4, 0.1% Triton X-100), and once in low salt (5 mM Tris-HCl, pH 7.4, 0.1% Triton X-100) wash buffers. Biotinylated RNA was extracted from the beads and precipitated in ethanol. 3′ RNA adaptors were ligated to biotinylated RNA and a second round of biotin-streptavidin purification was performed. The mRNA cap was then removed and the reverse 5′ RNA adaptor ligated. After the third round of biotin-streptavidin purification, adaptor ligated nascent RNA was reverse transcribed (RT) into complementary DNA (cDNA) using RP1 primer. cDNA was amplified with index primers and amplicons of 140–350 bp were selected using the Pippen Prep (Sage Science, Software: v.5.8) instrument. Equimolar concentrations of library fractions were then pooled together and sequenced using a high-output flow cell on the Illlumina NextSeq 500 platform. Raw reads from sequencing were demultiplexed allowing up to one mismatch using Illumina bcl2fastq2 v2.18. The adaptor sequence was removed, and reads were trimmed to 36 bp. After reverse complementing, reads were aligned to the dm6-hg38 combined genome using Bowtie2 (version 2.2.4) with default settings. The last base pair of the reads was used to generate BigWig files for visualization in a genome browser and for downstream analyses.

**Immunostaining.** HeLa cells grown on glass coverslips were fixed for 15 min at room temperature with 2% paraformaldehyde. Then, the cells were incubated for 15 min with PBST containing PBS and 0.5% Triton X-100. After blocking cells with PBS containing 10% normal goat serum, they were incubated at room temperature with primary antibodies to coilin (ab11822; Abcam) at 1:2000 dilutions, MED26 (D4B1, 14950S; Cell Signaling Technology) at 1:500 dilution, MED6 (sc-9433; Santa Cruz Biotechnology) at 1:100 dilution, MED24 (C-16, sc5338; Santa Cruz Biotechnology) at 1:100 dilution, NELFb (D6K9A, #14894; Cell Signaling Technology) at 1:100 dilution, CSTF64 (A301-092A; Bethyl Laboratories) at 1:100 dilution; AFF4 (A302-539A; Bethyl Laboratories) at 1:100 dilution; and ICE1 (KIAA0947) (A304-276A; Bethyl Laboratories) at 1:200 dilution in PBST containing 0.1% bovine serum albumin. The cells were then incubated with Alexa 488-labeled goat polyclonal antibody to mouse IgG at 1:1000 dilution, Alexa 555-labeled goat polyclonal antibody to rabbit IgG at 1:1000 dilution or Alexa 555-labeled rabbit polyclonal antibody to goat IgG at 1:1000 dilution (Life Technologies), covered with a drop of Prolong Gold antifade reagent (Invitrogen), and then photographed with a ZEISS LSM 700 Laser Scanning Microscope. Three-dimensional super-resolution images were acquired using a Leica TCS SP8 STED 3X Gated 660 system with a ×100 objective lens (HC PL APO CS2 ×100/1.40 NA OIL). The excitation was provided by a white light laser and the depletion was from a 660 nm STED laser with the three-dimensional slider adjusted to 60%, and the fluorescence signal was acquired using a Leica HyDTM in time-gated mode[39]. All images were deconvolved and arranged using the Huygens software (Scientific Volume Imaging B.V., The Netherlands) and Photoshop (Adobe, USA), respectively. Quantification of the number and signal intensity of the MED26 and ICE1 particles was performed using the ImageJ Fiji software.

**Construction of plasmids containing the HIST1H1C gene.** A plasmid containing the HIST1H1C gene with the HIST1H1C promoter was generated as follows. We defined minus (−) as before the TSS and plus (+) as after the TSS. The TSS was assigned the position of +1. The HIST1H1C gene region comprising the promoter (from −801 to −1), HIST1H1C gene region (from +1(TSS) to +762(TES)), and downstream region of the gene (from +763 to +881) containing the stem loop and PAS was amplified by PCR using the genome of HEK293T cells as a template. To distinguish the protein from endogenous histone protein, a FLAG-tag sequence was added to the end of the HIST1H1C coding region by PCR and subcloned into pBluescript II KS (Supplementary Fig. 9a). A plasmid containing the HIST1H1C gene with the CMV promoter was generated as follows. The HIST1H1C gene region comprising the HIST1H1C gene region (from +1(TSS) to +762(TES)) and downstream region of the gene (from +763 to +881) was amplified by genomic PCR. To distinguish the protein from endogenous histone protein, a FLAG-tag sequence was added to the end of the HIST1H1C coding region by PCR and subcloned into the CMV promoter-containing pcDNA3.1 hygro (−) (Supplementary Fig. 9a).

**Reporting summary.** Further information on research design is available in the Nature Research Reporting Summary linked to this article.

## Data availability

ChIP-seq, RNA-seq, and PRO-seq data for are deposited in GEO under accession GSE121024. The mass spectrometric datasets have been deposited to the ProteomeXchange via the MassIVE repository MSV000083465 [ftp://massive.ucsd.edu/MSV000083465]. Original data underlying parts of this study performed at the Stowers Institute can be downloaded from the Stowers Original Data Repository (http://www.stowers.org/research/publications/LIBPB-1361). Source data are provided as a Source Data file.

## Code availability

Mass spectrometry analysis tool called *swallow* (v. 0.0.1) is available at https://github.com/tzw-wen/kite.

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

## Acknowledgements

We thank Miho Uchiumi for help in preparing the manuscript, Hitomi Kurosawa, Nozomi Sakurai, Ami Kosaka, Takahiro Asanuma, and Naoki Takahashi for experimental assistance. We also thank J. Ludovic Croxford, Ph.D., of Edanz Group (www.edanzediting.com/ac) for editing a draft of this manuscript. This work was partly supported by Mizuho Oda and Emiko Koda in the Cooperative Research Project Program of the Medical Institute of Bioregulation, Kyushu University. This work was supported in part by KAKENHI (15H04690, 18H02607 to S.H., 15H04701, 16H06279, 17K19578, 18H02378, 19K22401, 221S0002, and 16H06279 to H.T.) and by Takeda Science Foundation (H.T.), Suhara Memorial Fund (H.T.), Takamatsu Cancer Research Fund (H.T.), The Ichiro Kanehara Foundation (H.T.), Friends of Leukemia Research Fund (H.T.), Ono Cancer Research Fund (H.T.), Kobayashi Foundation for Cancer Research (H.T.), MSD Life Science Foundation (H.T.), The Naito Foundation (H.T.), The Tokyo Biochemical Research Foundation (H.T.), Yokohama Foundation for Advancement of Medical Science (H.T.), The Uehara Memorial Foundation (H.T.), and by a grant from the Helen Nelson Medical Research Fund to the Stowers Institute (J.W.C.).

## Author contributions

H.T. planned the research, performed most of the experiments, and writing the manuscript. A.R. performed PRO-seq. K.C. generated and K.C., S.S., C.S., K.S., and Y.H. characterized mutant MED26 cells. H.S. performed ChIP assay. M.S., S.C., I.T., S.F., R.A., and M.I. analyzed the data from next-generation sequencing. A.S., Y.Z., L.F., M.P.W., M.M., and K.I.N. performed mass spectrometry analysis. Y.S., K.C., and A.R. performed ChIP-seq and RNA-seq analyses. H.H. performed immunostaining and T.H. performed the observation with STED microscopy. J.Y. and Y.Y. helped with interpretation of results. T.T., M.W., J.W.C., R.C.C., and S.H. contributed to the writing of the manuscript. H.T., S.H., and J.W.C. supervised the research.

## Competing interests

The authors declare no competing interests.
