## [Peer Review File · Nature Communications]

Reviewers' comments:

Reviewer #1 (Remarks to the Author):

In the manuscript "The role of Mediator and Little Elongation Complex in transcription termination" the authors make use of proteomic analysis and advanced microscopy to describe the role of mediator and little elongation complex in the transcription termination of non-polyadenylated genes at Cajal bodies.

With reference to the part of the manuscript supported by imaging methods:

The authors make use of confocal microscopy to study nuclear spatial localization of some mediator components (MED26, MED24 and MED6), a LEC component (ICE1), a NELF component (NELFb), a HLF component (CSTF64), a SEC component (AFF4) with a molecular marker of Cajal bodies, i.e. coilin.

Then they use STED super-resolution microscopy to study colocalization of the first 4 markers with coilin inside the cajal bodies. They are able to show a finer bead-like substructure within the Cajal bodies. This type of substructures is shared by Mediator and LEC markers, suggesting their synergistic role in transcription termination at Cajal bodies. The use of super-resolution microscopy is a technical aspect that improves the quality of the paper and points the direction for future investigations at the level of single Cajal bodies.

The labeling/imaging methods are described clearly.

I would like the authors to address the following concerns.

Major concerns

1) In figure 7a-d the authors study colocalization of different markers by confocal microscopy.

I have a major concern about the AFF4 staining reported in Fig.7g. As it appears from this figure, the staining is mainly cytoplasmic with no or little nuclear signal.

This data supports their conclusion that SEC is not colocalized with Cajal bodies: "but not AFF4, which is an SEC component (Fig. 7a–g). Thus, MED26-containing Mediator, LEC, NELF and HLF, but not SEC, are colocalized at Cajal bodies."

If the authors do not see any expression in the nucleus, how could they talk about nuclear colocalization?

According to literature, AFF4 should be expressed in the nucleus, and appears as nuclear speckles.

For instance see images of AFF4 nuclear speckles reported in: Melko et al, Functional characterization of the AFF (AF4/FMR2) family... Hum Mol Genet. 2011.

It is very common to get wrong staining (e.g. cytoplasmic rather nuclear) depending on the antibodies or the sample preparation/labeling protocol.

In the way they are presented, the data on AFF4 are not convincing.

2) In figure 7i-k the authors study nanoscale spatial distribution of different markers by STED microscopy. These data are quite interesting as they show finer details of colocalization at the level of Cajal bodies.

Besides the purely visual representation of colocalization, the authors should provide some more quantitative analysis of the mutual relationship between the markers.

At minimum, I suggest they plot line intensity profiles across the Cajal bodies. The line intensity profiles could help the reader to better appreciate the 'granular' or 'bead-like' features described by the authors and get an idea of the typical size of these features. Moreover, it could also help the reader to observe colocalization between the different markers and coilin.

Reviewer #2 (Remarks to the Author):

The manuscript by Takahashi et al. 'The role of mediator and Little Elongation Complex (LEC) in transcription termination' provides an important insight into the molecular components and

potential understanding of how specific transcription complexes interact to regulate transcription termination of snRNA genes and replication dependent histone (RDH) genes. The authors use a range of established methods to establish the requirement for LEC in non-poly(A) transcription termination. LEC was shown to co-purify with proteins required for the non-poly(A) termination of specific genes in particular snRNA and RDH genes.

These experiments outline a role for specific LEC components in the termination of RDH genes. The findings are relevant for understanding the relationship between mediator and transcriptional termination of snRNA and RDH genes and point to unique roles for LEC protein components in RDH gene regulation. Nevertheless, the manuscript falls short of providing compelling evidence that demonstrates a significant step in understanding the transcriptional consequences of the proposed mechanism termination at poly(A) versus upstream termination sites. Data pointing to the functional consequence of failure to terminate transcription at non-poly(A) sites would add considerably to this manuscript.

Comments

1) What are the functional consequences of transcription termination producing polyadenylated mRNAs of genes that would otherwise produce a stem-loop structure instead of poly(A) tails? Could the authors speculate why cells evolved to terminate these genes upstream of polyA sites? Does it affect cell cycle, nucleosome structure, chromatin remodelling? Replication dependent Histone genes are expressed predominantly through S-phase and supply cells with requisite histones necessary for the DNA replication. Changes Histone transcripts may translate into changes in nucleosome structure which can have profound effects: transcription kinetics, the association of key regulatory proteins, the association with chromatin remodelling complexes. The polyadenylation sites downstream of the of HDEs are suggested to act as a mechanism to prevent PolII from reading through into regions downstream of the RDH genes. Is this the likely the only role of these downstream polyA sites?

2) Is this a widespread mechanism? Similar and/or supporting data using other cell lines would be of enormous benefit. Equally how evolutionary conserved would this process be? Can the authors comment from literature on similar observations and mechanisms from yeast, flies or worm models?

3) The authors performed RNA seq from polyA-selected libraries and ribo depleted libraries prepared from HEK293T cells. RNA seq of poly(A) mRNA libraries to show enrichment of genes upregulated in cells in which MED26 transcripts have been reduced (siRNA). RDH genes were among those genes upregulated when MED26 levels were reduced. Complementary RNA seq from ribo-depleted libraries showed either no change or even a reduction in transcripts that show increases from the poly A library analysis. Regarding the later the authors comment on the unexpected finding that those transcripts upregulated in the polyA libraries were either unchanged or reduced. Could the authors comment further? I agree the reduction of transcripts is unexpected however one would have expected to see no difference between control and MED26siRNA treated cells.

4) Immuno-precipitation of the complexes using epitope (FLAG)-tagged proteins is powerful and has been exploited extensively by these authors in the past, they are clearly well positioned to interrogate this kind data with caution. Nevertheless, proteomic analyses that affinity purify endogenous proteins would eliminate artefacts induced by expression of epitope tagged proteins versions. For example, the authors affinity purify FLAG-tagged ICE 1 from 293 cells and identify interacting proteins that are enriched for the pull down with anti-flag antibodies. Interacting proteins from mediator were shown (Fig 5) however MED26 was absent from this analysis? Given the preceding data focus on the interaction with MED26 and point to the NTD of MED 26 as the domain responsible for the recruitment of factors necessary to carry out 3'-termination could the authors comment on the absence of MED26 in this analysis? The authors use Ab's raised against endogenous versions of MED26, ICE1, ELL, CPSF73 etc in ChIP assays, have the authors considered proteomic analyses on these rather than FLAG-tagged versions?

5) The authors have published many proteomics data sets using NSAF as a method of providing relative quantitation in their pull-down experiments. I have few issues with spectral counting as a means of relative quantification and the authors are experts, it nevertheless remains a rather

imprecise method for protein quantification. The field of relative proteomic quantification is improving rapidly both in technical developments at an instrument level and in software used for analysis. Could the authors provide details of the instrumentation and methods used to generate this data. The references point to two publications from 2001 and 2006, an update in the main body of text or supplementary section would be appreciated.

6) How does NSAF compare with rapidly evolving quantitative methods such as isobaric labelling using tandem mass tags and where multiplex capacity now currently stands at 11 samples in a single run. The statistical power, assessment of variability, ability to generate PCA analysis and assess biological vs batch effects is considerable using such methods. Furthermore, potentially unlimited comparisons can be now achieved by the inclusion of reference channels in the experiments. Could data regarding replicates (biological and technical), statistics/variability be included in the main body of text or supplementary materials.

7) The software used (DTASelect and CONTRAST) were first published in 2002, what versions were used for these analyses? Other software eg SAINT may provide better precision and alternatively, as alluded to above, isotopic labelling methods such as SILAC or isobaric labelling would provide better quantitative accuracy with fewer missing values that could improve the sensitivity and quantitative accuracy of the data.

Reviewer #3 (Remarks to the Author):

In their manuscript entitled "The role of Mediator and Little Elongation Complex in transcription termination", Takahashi et al report that the Mediator component, Med26, recruits the Little Elongation Complex (LEC) to replication dependent histone genes (RDH) and regulates their 3' end processing. The LEC is shown to exist in a complex with both 5' cap-binding factors and 3' end processing factors and to localize to Cajal bodies. Depletion or mutation of Med26 or depletion of LEC results in alterations in expression of both RDH and snRNA genes.

The current extend previous observations from this group and others. It has been shown previously that the LEC, which is recruited to snRNA genes by Med26, regulates their initiation and elongation. It was also known that 3' end processing of snRNA transcripts is affected by factors that regulate their transcription: replacing an snRNA promoter with an actin promoter in drosophila alters 3' end processing. Thus, the association between recruitment of transcription initiation factors, such as LEC, and 3' processing factors was already predicted. The localization of Med26 and ICE to Cajal bodies was also reported. The novel aspects of this study are: 1) the demonstration that the LEC is recruited to RDH genes, 2) 3' processing of RDH genes is LEC dependent and 3) the LEC is in a complex with 5' cap-binding factors and 3' end processing factors. The authors have provided extensive evidence in support of their model that Med26 recruits the LEC to RDH and snRNA gene promoters, which in turn brings in the 3' processing factors; in the absence of Med26 or LEC, transcription reads through the normal RDH termination sites, leading to aberrant polyA+ addition to the RDH. Overall, the studies are convincing. However, the authors need to address a number of issues, as detailed below.

Fig. 1 –

- The depletion of Med26 results in an increase in unprocessed transcripts which is statistically significant. However, the actual fraction of UT, relative to CT, is extremely small, less than 1% in many cases. The increases are even less. As reported previously, Med26 results in decreased gene expression. Could those small variations reflect non-specific effects? Internal controls need to be provided, such as the effect on read-through of non-LEC genes, to demonstrate specificity of the effect.

Fig. 2 –

- Knock-down (KD) of ICE1 and Med26, but not the SEC component AFF4, increases UT/CT ratio. The authors argue that this suggests that Med26 recruits LEC which regulates 3' termination. Why is the effect of ICE1 less than Med26 in all cases? Since Med26 also recruits SEC, what is the effect of Med26 KD on read-through of an SEC dependent gene (which is a different question than the effect of AFF4 on an RDH gene).

- KD of ZC3H8 also increases the UT/CT ratio, but except for H1H3E, the ratios are less than 0.1% . Even if statistically significant, what is the biological significance of such a small increase in polyA+histone mRNA? This needs to be discussed in terms of the overall biological significance of the LEC control of 3' end processing.
- In Fig. 2c,d the Venn diagram and table presumably summarize the status of polyA+ RDH RNA following KD. However, neither the figure nor the legend so indicate. If this is not the case, the results are inconsistent with previous results.
- The majority of ICE1 regulated genes that show an increase in polyA+RNA do they not overlap with Med26. Why?

Fig. 3 -

- The Figure shows that published ChIP-seq data of ELL and ZC3H8 overlap with Med26 ChIPseq, consistent with Med26 recruiting the LEC to RDH genes. However, the number of genes associated with ZC3H8 is much greater than with ELL. Since ELL is in both SEC and LEC, while ZC3H8 is only in LEC, why are the number of genes associated with ZC3H8 so much greater? Why isn't there a greater overlap of Med26 with ELL and ZC3H8? The authors need to discuss these points.

Fig. 4-

- The authors conclude from Fig. 4 that the Med26 mutant enhances read-through. However, the metadata analysis in fig. 4D, clearly shows that the read counts in the gene body are also increased such that the overall distribution of reads does not appear to be different between WT and Mut in histone genes. This increase is clearly apparent in browser views in a) and c). The UT/CT ratio, based on number of reads, should be calculated from the metadata to determine whether there is a significant difference between WT and mutant Med26. The CT calculation should be based on reads downstream of the promoter, not upstream (see below).
- In the supplementary figure, a metadata analysis of UT/CT has been done. However, the ratio determined was comparing reads 100 bp upstream to the downstream reads. This would not distinguish between increases of readthrough alone from increases across the gene body. The data should be recalculated comparing reads 50bp downstream of the promoter with readthrough.
- snRNA genes more clearly show increased readthrough. However, the meta
- effects are modest. The analysis should be of polyA+ histones, as in Fig. 1.

Fig. 5-

- Both pull-down gel and mass spec analyses document that ICE1 is recovered in a complex with other LEC, CBCA-NELF and termination factors. However, it is surprising that Med26 is not recovered in either the pull-down gel or mass spec. This brings into question the entire premise of the study that Med26 recruits the LEC. The authors need to explain this.
- NSAF is defined in the supplementary figure, but it should be defined in the legend to fig. 5.

Fig. 8 -

- The role of ICE1 in recruiting HLF and Integrator is assessed by determining the effect of ICE1 knock-down on CPSF73 and Ints9, respectively. However, neither of those proteins appears in the mass spec results shown in Fig. 5. Therefore, the authors need to explain how those two proteins were chosen. Were others that did show up in the mass spec tested?

In conclusion, the authors report an interesting study in which 3' processing factors and cap-binding factors are in a complex with the LEC, which is recruited to RDH and snRNA genes by Med26, resulting in proper 3' end processing. However, to firmly document their model, the authors need to address two issues: 1) is the observed increased read-through with Med26 KD due to defective 3' end processing or to increases across the gene body which spill over to the 3' end and 2) what is the effect of replacing an RDH promoter with a promoter dependent on the SEC, to determine the effect on 3' processing.

Reviewer #4 (Remarks to the Author):

In this report, Takahashi et al reported that Med26 and the Little Elongation Complex (LEC)

function in replication-dependent histone (RDH) and snRNA transcription termination. Mechanistically the authors provided evidence that the Med26 recruits LEC, which forms a large complex with various termination factors. Overall the study is well performed and presents potentially novel insights into RDH and snRNA termination as well as Mediator and LEC functions. However, there are some technical issues and their model is not fully substantiated by the data:

1. The authors proposed a model in which Med26 recruits LEC to RDH and snRNA genes. However, Flag-ICE1 immunoprecipitation failed to co-precipitate Med26 while Med31 was detected. The authors need to reconcile this with their model.
2. Instead of Med26 recruits heat labile factors to promote processing of RHD transcripts to form unpolyadenylated mRNAs, is it possible that Med26 inhibits the cleavage/polyadenylation pathway?
3. Knockdown of ICE1 led to a decrease in the recruitment of CPSF73 and Ints9 (Fig. 8) and the authors argued that this is the evidence that LEC helps to recruit termination factors. Does Med26 knockdown also have the same effect? Does ICE1 knockdown lead to a decrease in PolII density? If so, could the decrease in CPSF or Integrator recruitment be an indirect effect of lower PolII activity?
4. In most studies, Med26 knockdown was performed. In assessing the effect on transcription, however, a hypomorphic mutant was used. What is the rationale for this? Is there a difference in the results between a knockdown and this mutant?

The reviewers' comments are listed below in boldface black characters, and our point-by-point responses to the comments are shown in regular green characters.

Reviewers' comments:

Reviewer #1 (Remarks to the Author):

In the manuscript “The role of Mediator and Little Elongation Complex in transcription termination” the authors make use of proteomic analysis and advanced microscopy to describe the role of mediator and little elongation complex in the transcription termination of non-polyadenylated genes at Cajal bodies. With reference to the part of the manuscript supported by imaging methods: The authors make use of confocal microscopy to study nuclear spatial localization of some mediator components (MED26, MED24 and MED6), a LEC component (ICE1), a NELF component (NELFb), a HLF component (CSTF64), a SEC component (AFF4) with a molecular marker of Cajal bodies, i.e. coilin. Then, they use STED super-resolution microscopy to study colocalization of the first 4 markers with coilin inside the cajal bodies. They are able to show a finer bead-like substructure within the Cajal bodies. This type of substructures is shared by Mediator and LEC markers, suggesting their synergistic role in transcription termination at Cajal bodies. The use of super-resolution microscopy is a technical aspect that improves the quality of the paper and points the direction for future investigations at the level of single Cajal bodies. The labeling/imaging methods are described clearly.

I would like the authors to address the following concerns.

Major concerns

1) In figure 7a-d the authors study colocalization of different markers by confocal microscopy. I have a major concern about the AFF4 staining reported in Fig.7g. As it appears from this figure, the staining is mainly cytoplasmic with no or little nuclear signal. This data supports their conclusion that SEC is not colocalized with Cajal bodies: “but not AFF4, which is an SEC component (Fig. 7a–g). Thus, MED26-containing Mediator, LEC, NELF and HLF, but not SEC, are colocalized at Cajal bodies.” If the authors do not see any expression in the nucleus, how could they talk about nuclear colocalization? According to literature, AFF4 should be

expressed in the nucleus, and appears as nuclear speckles. For instance, see images of AFF4 nuclear speckles reported in: Melko et al, Functional characterization of the AFF (AF4/FMR2) family... Hum Mol Genet. 2011. It is very common to get wrong staining (e.g. cytoplasmic rather nuclear) depending on the antibodies or the sample preparation/labeling protocol. In the way they are presented, the data on AFF4 are not convincing.

We agree with the reviewer and have added new results for immunostaining of AFF4 obtained using a different anti-AFF4 antibody. As shown in the new Fig. 7g, AFF4 immunostaining was detected in the nucleus and not colocalized with coilin at Cajal bodies. These findings are consistent with our model that MED26 and LEC, but not AFF4, are specifically colocalized at Cajal bodies. (Page 18, lines 16–18)

2) In figure 7i-k the authors study nanoscale spatial distribution of different markers by STED microscopy. These data are quite interesting as they show finer details of colocalization at the level of Cajal bodies. Besides the purely visual representation of colocalization, the authors should provide some more quantitative analysis of the mutual relationship between the markers.

We thank the reviewer for this comment. To address this suggestion, we quantified the number of particles of ICE1 and MED26 and calculated the intensity of the particles in nuclei of HeLa cells. As shown in the new Fig. 7h, we observed nuclei of 33 HeLa cells and found that 43 of 132 ICE1 particles were colocalized with coilin at Cajal bodies and that the size of ICE1 particles colocalized with coilin was larger than that of 89 particles not colocalized with coilin. In addition, the signal intensity of ICE1 particles colocalized with coilin was much higher than that of particles not colocalized with coilin. Furthermore, as shown in the new Fig. 7i, we found that 59 of 176 MED26 particles in 18 HeLa cell nuclei colocalized with coilin and that the size of MED26 particles colocalized with coilin was larger than that of 117 MED26 particles not colocalized with coilin. In contrast to the ICE1 particles, the intensity of MED26 particles colocalized with coilin was similar to that of MED26 particles not colocalized with coilin. In light of our previous evidence that MED26 plays a role in transcriptional regulation of SEC-targeted genes that encode polyadenylated transcripts, including *c-myc*, in addition to LEC-targeted genes encoding non-polyadenylated transcripts, it is possible that the MED26 particles not colocalized with coilin are involved in transcriptional regulation of SEC-targeted genes. (Page 18, lines 22– Page 19, lines 16)

At minimum, I suggest they plot line intensity profiles across the Cajal bodies. The line intensity profiles could help the reader to better appreciate the ‘granular’ or ‘bead-like’ features described by the authors and get an idea of the typical size of these features. Moreover, it could also help the reader to observe colocalization between the different markers and coilin.

In the revised manuscript we present line scan analyses in which we plot the line intensity across the Cajal bodies (new Fig. 8e–h). As the reviewer suggested, these plots indeed reinforce the idea that Mediator subunits including MED26, MED24, and MED6, and coilin form granular or bead-like structures at Cajal bodies. (Page 19, lines 22–23), and they help document colocalization between coilin and the various markers.

Reviewer #2 (Remarks to the Author):

The manuscript by Takahashi et al. ‘The role of mediator and Little Elongation Complex (LEC) in transcription termination’ provides an important insight into the molecular components and potential understanding of how specific transcription complexes interact to regulate transcription termination of snRNA genes and replication dependent histone (RDH) genes. The authors use a range of established methods to establish the requirement for LEC in non-poly(A) transcription termination. LEC was shown to co-purify with proteins required for the non-poly(A) termination of specific genes in particular snRNA and RDH genes. These experiments outline a role for specific LEC components in the termination of RDH genes. The findings are relevant for understanding the relationship between Mediator and transcriptional termination of snRNA and RDH genes and point to unique roles for LEC protein components in RDH gene regulation. Nevertheless, the manuscript falls short of providing compelling evidence that demonstrates a significant step in understanding the transcriptional consequences of the proposed mechanism termination at poly(A) versus upstream termination sites. Data pointing to the functional consequence of failure to terminate transcription at non-poly(A) sites would add considerably to this manuscript.

Comments

1) What are the functional consequences of transcription termination producing

polyadenylated mRNAs of genes that would otherwise produce a stem-loop structure instead of poly(A) tails? Could the authors speculate why cells evolved to terminate these genes upstream of polyA sites? Does it affect cell cycle, nucleosome structure, chromatin remodeling? Replication dependent Histone genes are expressed predominantly through S-phase and supply cells with requisite histones necessary for the DNA replication. Changes in Histone transcripts may translate into changes in nucleosome structure which can have profound effects: transcription kinetics, the association of key regulatory proteins, the association with chromatin remodeling complexes. The polyadenylation sites downstream of the of HDEs are suggested to act as a mechanism to prevent Pol II from reading through into regions downstream of the RDH genes. Is this the likely the only role of these downstream polyA sites?

We agree with the reviewer that future studies addressing the functional consequences of changing the balance between non-polyadenylated and polyadenylated forms of RDH transcripts will be of great interest. However, as the reviewer points out these are likely to be complex, and unraveling these consequences is, we believe, beyond the scope of the present study, which defines new and unanticipated roles of the Mediator and LEC in 3'-processing and termination of non-polyadenylated transcripts.

As the reviewer notes and as we point out in the manuscript (Page 4, lines 21–Page 5, lines 10), one proposed function for polyadenylation of RDH transcripts is to provide a fail-safe mechanism to ensure that Pol II does not read-through into downstream regions. This is potentially important, since RDH loci are gene-dense regions with ample opportunity for read-through transcription from one RDH gene to alter or interfere with expression of nearby genes, including other RDH genes.

Of note, and as also pointed out in the text of both the original and revised manuscript (Page 5, lines 10–12), recent evidence also supports the idea that polyadenylated RDH transcripts contribute to low-level expression of replication-dependent histones outside of S phase and in long lived, terminally differentiated cells, where they have been proposed to serve as a source of replacement histones¹. Hence, it is tempting to speculate that Mediator-dependent changes in the balance between non-polyadenylated and polyadenylated forms of RDH transcripts could contribute to the decision to synthesize polyadenylated RDH transcripts in terminally differentiated tissues.

Finally, there is substantial evidence that excessive accumulation of replication-dependent histones outside of S-phase is toxic, and use of non-polyadenylated

transcripts is a key mechanism for ensuring that the high levels of Histone mRNAs needed during S-phase disappear during other phases of the cell cycle. Non-polyadenylated Histone mRNAs are only stable during S-phase, because stem-loop binding protein (SLBP), which is only present during S-phase, binds to the conserved stem loop structure of RDH transcripts to prevent non-polyadenylated RDH mRNAs from undergoing degradation during S-phase². At the end of S-phase, SLBP is phosphorylated by cyclin-dependent kinase and degraded by the ubiquitin proteasome pathway. Degradation of SLBP outside of S-phase is tightly associated with disappearance of Histone mRNAs outside of S-phase³. Thus, SLBP plays an important role in ensuring high levels of Histone mRNAs and Histone proteins at S-phase to prevent harmful accumulation of free Histone proteins in cells outside of S-phase^{2, 4, 5, 6}. Considering that SLBP also copurified with FLAG-tagged ICE1 (Revised Fig. 6a) and was previously identified as a component of CBCA-NELF-DSIF⁷, our results are consistent with the possibility that in proliferating cells, LEC recruited by MED26-containing Mediator to RDH genes helps to restrict synthesis of RDH transcripts to prevent harmful production of free Histone proteins in cells outside of S-phase. We comment on this possibility in the Discussion section (Page 27, lines 6–22).

2) Is this a widespread mechanism? Similar and/or supporting data using other cell lines would be of enormous benefit. Equally how evolutionary conserved would this process be? Can the authors comment from literature on similar observations and mechanisms from yeast, flies or worm models?

We appreciate the reviewer for raising this important point. In accordance with the reviewer's suggestion, we examined whether transcription termination regulation of RDH genes is regulated by MED26 in other cell lines. For this, we generated an HCT116 cell line with the Auxin Inducible Degron (AID) tag sequence inserted at the 3'-end of the coding sequence of the endogenous MED26 gene to drive expression of MED26 with a C-terminal auxin-inducible degron (MED26-AID). As shown in new Fig. 1d, treatment of cells with auxin, leading to acute loss of MED26, resulted in an increase in the ratio of polyadenylated RDH transcripts relative to total transcripts, similar to what was seen in HEK293 cells after siRNA-mediated MED26 knockdown. In addition, auxin treatment led to a significant increase in levels of unprocessed RDH transcripts in MED26-AID-expressing HCT116 cells, but not in parental HCT116 cells (new Fig. 1e). This result suggests that regulation of 3'-processing and termination by MED26 and LEC is at least conserved in other types of human cells. (Page 10, lines 5–

15)

Furthermore, MED26 is a metazoan-specific subunit in the Mediator complex and conserved from *Drosophila melanogaster* to humans, but does not exist in yeast, indicating that the machinery is metazoan-specific mechanisms. In accordance with the reviewer's suggestion, we have mentioned this issue in the Discussion section and stated: "Considering that MED26 is a metazoan-specific subunit of the Mediator complex and conserved from *Drosophila melanogaster* to humans, but is not present in yeast, we expect that the mechanism uncovered in our study machinery is metazoan-specific." (Page 25, lines 3–5)

3) The authors performed RNA-seq from polyA-selected libraries and ribo depleted libraries prepared from HEK293T cells. RNA-seq of poly(A) mRNA libraries to show enrichment of genes upregulated in cells in which MED26 transcripts have been reduced (siRNA). RDH genes were among those genes upregulated when MED26 levels were reduced. Complementary RNA-seq from ribo-depleted libraries showed either no change or even a reduction in transcripts that show increases from the poly A library analysis. Regarding the later the authors comment on the unexpected finding that those transcripts upregulated in the polyA libraries were either unchanged or reduced. Could the authors comment further? I agree the reduction of transcripts is unexpected however one would have expected to see no difference between control and MED26 siRNA treated cells.

We appreciate the reviewer for raising this issue. As the reviewer pointed out, RDH transcripts upregulated in polyA-selected libraries were either unchanged or reduced in ribo-depleted libraries following knockdown of MED26 (new Fig. 1a). We also found that majority of RDH transcripts upregulated in polyA-selected libraries were reduced in ribo-depleted libraries following acute loss of MED26 in MED26-AID expressing cell line or in MED26 hypomorphic mutant cells (new Fig. 1d, new Fig. S5a). In addition, we found that induction of RDH genes after release of the cells from S-phase was decreased in MED26 hypomorphic mutant cells (new Fig. S5c). These results suggest that MED26 contributes to multiple processes of RDH gene expression, likely including not only transcription termination and 3'-end processing, but also transcription initiation and elongation. We now make this point explicitly on Page 12, lines 10–Page 13, lines 7.

4) Immuno-precipitation of the complexes using epitope (FLAG)-tagged proteins is powerful and has been exploited extensively by these authors in the past, they are clearly well positioned to interrogate this kind data with caution. Nevertheless, proteomic analyses that affinity purify endogenous proteins would eliminate artefacts induced by expression of epitope tagged proteins versions. For example, the authors affinity purify FLAG-tagged ICE1 from 293 cells and identify interacting proteins that are enriched for the pull down with anti-FLAG antibodies. Interacting proteins from Mediator were shown (Fig 5) however MED26 was absent from this analysis? Given the preceding data focus on the interaction with MED26 and point to the NTD of MED26 as the domain responsible for the recruitment of factors necessary to carry out 3'-termination could the authors comment on the absence of MED26 in this analysis?

We appreciate the reviewer for raising this important point. We agree with the reviewer and have added the results of new experiments showing that FLAG-tagged ICE1 (F-ICE1) copurified with MED26-containing Mediator. We purified F-ICE1 from 293FRT cells stably expressing F-ICE1 and performed western blot analyses. As shown in the new Fig. 6b, F-ICE1 copurified with Mediator subunits including MED26, MED1, and MED23, while F-BTBD19 did not. Of note, both F-ICE1 and F-AF9 copurified with Mediator subunits including MED26 and MED1 (Revised Fig. 6a), consistent with previous results showing that MED26 interacts with SEC^{8,9}. These results demonstrate that ICE1 interacts with MED26-containing Mediator. As the reviewer pointed out, we did not detect MED26 in F-ICE1 immunoprecipitates in our proteomics analysis (Fig. 5b and revised Fig. 5c). Because protein modifications, especially phosphorylation, can interfere with ionization of peptides and identification of proteins by mass spectrometry, our results raise the possibility that protein modifications such as phosphorylation make it difficult to detect MED26 in mass spectrometric analysis.

5) The authors use Ab's raised against endogenous versions of MED26, ICE1, ELL, CPSF73 etc in ChIP assays, have the authors considered proteomic analyses on these rather than FLAG-tagged versions?

We appreciate the reviewer's suggestion. In general, proteomics analysis of immunoprecipitates using antibodies against endogenous proteins can be technically difficult, because large amounts of immunoglobulins derived from antibodies are included in the immunoprecipitates throughout the elution process and interfere with

identification of other proteins by mass spectrometry. Instead of mass spectrometric analysis, we immunopurified endogenous ICE1 from nuclear extracts of HEK293T cells using two kinds of affinity-purified ICE1 antibodies and performed western blot analyses. As shown in the new Fig. 6c, in experiments performed with both ICE1 antibodies, endogenous ICE1 co-purified with NELFb, Symplekin, and INTS4, indicating that endogenous ICE1 interacts with the components of NELF, Heat labile factor (HLF), and Integrator. These results support our data for mass spectrometric analysis of FLAG-tagged ICE1-interacting proteins showing that ICE1 interacts with NELF, HLF, and Integrator (Fig. 5b and revised Fig. 5c). (Page 17, lines 15–20).

6) The authors have published many proteomics data sets using NSAF as a method of providing relative quantitation in their pull-down experiments. I have few issues with spectral counting as a means of relative quantification and the authors are experts, it nevertheless remains a rather imprecise method for protein quantification. The field of relative proteomic quantification is improving rapidly both in technical developments at an instrument level and in software used for analysis. Could the authors provide details of the instrumentation and methods used to generate this data. The references point to two publications from 2001 and 2006, an update in the main body of text or supplementary section would be appreciated.

Additional information about the data acquisition and processing have been added to the “MudPIT analysis” section in the Methods in the revised manuscript. This section now reads as follows (Page 33, lines 15–Page 34, lines 18):

MudPIT analysis

FLAG-immunoaffinity (Fig. 5c) or glutathione-affinity (Supplemental Fig. S4) purified proteins and negative controls were treated with benzonase and TCA precipitated before analysis by multidimensional protein identification technology (MudPIT)^{61,62}. After denaturation, reduction, and alkylation, proteins were digested with endoproteinase LysC followed by trypsin (Promega). Peptides mixtures were analyzed through 10 multidimensional liquid chromatography steps implemented on a quaternary Agilent 1100 series HPLC in-line with a Thermo linear ion trap mass spectrometer. Tandem mass (MS/MS) spectra were interpreted using SEQUEST (v. 27)^{63,64} or ProLuCID (v. 1.3.3)⁶⁴, against databases of non-redundant human proteins (downloaded from NCBI on 6-10-2016 or 3-25-2015 for the data in Fig. 5c and Supplemental Fig. S4, respectively) and complemented with sequences from usual contaminants (human

keratins, IgGs, proteolytic enzymes). To estimate false positive discovery rates (FDRs), each sequence was randomized keeping amino acid composition and length the same, and the resulting ‘shuffled’ sequences were added to the ‘normal’ database (doubling its size) and searched at the same time. Peptide/spectrum matches were sorted and selected using DTASelect/CONTRAST (v. 1.9)⁶⁵ in combination with an in-house software, *swallow* (v. 0.0.1), to filter spectra, peptides, and proteins at FDRs <0.5%. Combining all runs, proteins had to be detected by at least 2 such peptides or by 1 peptide with 2 independent spectra. To estimate relative protein levels, distributed normalized spectral abundance factors (dNSAF) were calculated^{61,66,67,68}. The dNSAF for a protein *k* is proportional to the amount of the protein present in the sample and is calculated by the formula:

$$dNSAF_i = \frac{dSAF_i}{\sum_{i=1}^N dSAF_i}$$

$$dSAF_i = \frac{uSpC_i + \frac{uSpC_i}{\sum_{m=1}^M uSpC_m} \times sSpC_i}{Length_i}$$

With

in which shared spectral counts (sSpC) are distributed based on spectral counts unique to each protein *i* (uSpC) divided by the sum of all unique spectral counts for the *M* protein isoforms that shared peptide *j* with protein *i*.

7) How does NSAF compare with rapidly evolving quantitative methods such as isobaric labelling using tandem mass tags and where multiplex capacity now currently stands at 11 samples in a single run. The statistical power, assessment of variability, ability to generate PCA analysis and assess biological vs batch effects is considerable using such methods. Furthermore, potentially unlimited comparisons can be now achieved by the inclusion of reference channels in the experiments. Could data regarding replicates (biological and technical), statistics/variability be included in the main body of text or supplementary materials.

NSAF and its newest implementation dNSAF, which takes into account peptides shared between multiple proteins, have been used in hundreds of published manuscripts (from our group and others) as useful abundance measures for straightforward studies of

affinity purifications analyzed by mass spectrometry (APMS). In the present manuscript, proteins interacting with ICE1 were readily identified and quantified relatively to the bait protein by comparison against the proteins detected in FLAG negative controls (as shown in the expanded version of new Fig. 5c). Costly isobaric labeling, using tandem mass tags for example, is not necessary in cases such as this one, where we are not trying to accurately and precisely measure small changes in protein levels between samples or when multiplexing the LC/MS analysis is not necessary to reduce technical variations. In APMS studies, proteins that copurify with the bait should be among the most abundant proteins that are enriched relative to negative controls (i.e., in negative controls they are either not detected or fewer peptides/spectra are detected). Such proteins will be ranked as enriched by either label-free (dNSAF) or labeled (TMT) quantitation methods. In summary, the LC/MS analysis performed here didn't need to be quantitative to accomplish the goals of this work, namely, to identify potential interactors that we have followed up on and confirmed by other methods (see Western blots in new Fig. 6a–c).

8) The software used (DTASelect and CONTRAST) were first published in 2002, what versions were used for these analyses? Other software eg SAINT may provide better precision and alternatively, as alluded to above, isotopic labelling methods such SILAC or isobaric labelling would provide better quantitative accuracy with fewer missing values that could improve the sensitivity and quantitative accuracy of the data.

All software versions have now been included in the “MudPIT analysis” section of Material and Methods as shown in comment #6) above.

Reviewer #3 (Remarks to the Author):

In their manuscript entitled “The role of Mediator and Little Elongation Complex in transcription termination”, Takahashi et al report that the Mediator component, Med26, recruits the Little Elongation Complex (LEC) to replication dependent histone genes (RDH) and regulates their 3' end processing. The LEC is shown to exist in a complex with both 5' cap-binding factors and 3' end processing factors and to localize to Cajal bodies. Depletion or mutation of Med26 or depletion of LEC results in alterations in expression of both RDH and snRNA genes. The current study extends previous observations from this group and others. It has

been shown previously that the LEC, which is recruited to snRNA genes by Med26, regulates their initiation and elongation. It was also known that 3' end processing of snRNA transcripts is affected by factors that regulate their transcription: replacing an snRNA promoter with an actin promoter in drosophila alters 3' end processing. Thus, the association between recruitment of transcription initiation factors, such as LEC, and 3' processing factors was already predicted. The localization of Med26 and ICE to Cajal bodies was also reported. The novel aspects of this study are: 1) the demonstration that the LEC is recruited to RDH genes, 2) 3' processing of RDH genes is LEC dependent and 3) the LEC is in a complex with 5' cap-binding factors and 3' end processing factors.

The authors have provided extensive evidence in support of their model that Med26 recruits the LEC to RDH and snRNA gene promoters, which in turn brings in the 3' processing factors; in the absence of Med26 or LEC, transcription reads through the normal RDH termination sites, leading to aberrant polyA+ addition to the RDH. Overall, the studies are convincing. However, the authors need to address a number of issues, as detailed below.

Fig. 1 –

1) The depletion of Med26 results in an increase in unprocessed transcripts which is statistically significant. However, the actual fraction of UT, relative to CT, is extremely small, less than 1% in many cases. The increases are even less. As reported previously, Med26 results in decreased gene expression. Could those small variations reflect non-specific effects? Internal controls need to be provided, such as the effect on read-through of non-LEC genes, to demonstrate specificity of the effect.

We appreciate the reviewer's comment. Nevertheless, we think it is worth mentioning that while we agree that the fraction of UT is in many cases is very small even after MED26 knockdown or in MED26 mutant cells, in others, it can reach as high as 10-40% of CT after knockdown.

In accordance with the reviewer's suggestion, we tested the effect of MED26 knockdown on read-through (UT/CT ratio) of two genes that encode polyadenylated transcripts, *GAPDH* and the well-characterized SEC target, *c-myc*. As shown in the revised Fig. 1c and Fig. 2a, knockdown of MED26 did not affect the level of unprocessed, read-through transcripts at either *GAPDH* or *c-myc* (Page 10, lines 23–Page 11, line 3). Finally, as shown in new Fig. 4h and S6e, we observe that transcription

read-through at non-RDH protein-coding genes is the same in wild type and MED26 hypomorphic mutant cells when measured using PRO-seq or Pol II ChIP-seq read-through ratios. (Page 14, lines 1–13)

Fig. 2 –

2) Knock-down (KD) of ICE1 and Med26, but not the SEC component AFF4, increases UT/CT ratio. The authors argue that this suggests that Med26 recruits LEC which regulates 3' termination. Why is the effect of ICE1 less than Med26 in all cases?

We thank the reviewer for raising this issue. We agree with the reviewer's comments that the effect of ICE1 knockdown on the read-through transcripts of RDH genes is less than the effect of MED26 knockdown. There are several potential explanations for this observation. First, siRNA mediated knockdown may reduce levels of different targets to different extents. Although we have not performed rigorous quantitation of our western blots, it is worth noting that there appears to be more residual ICE1 than MED26 after siRNA treatment (Fig. S2, compare panels a and b). Second, Mediator could in principle affect processes other than LEC recruitment. To address this issue, we performed several new experiments. We found that depletion of MED26 or N-terminal deletion of MED26 decreased the total RDH transcripts in ribo-depleted RNA-seq libraries and the induction of RDH genes at S phase (Fig. 1a and d, Fig. S5a, Fig. S5c), while ICE1 knockdown did not affect total levels of RDH transcripts (new Fig. S7d). Consistent with these results, knockdown of MED26 decreased the occupancy of LEC component ELL, Pol II, and HLF component CPSF3 at RDH genes (Fig. 3d and new Fig. 9c). In contrast, knockdown of ICE1 decreased the occupancy of CPSF3 at RDH genes, but did not affect the occupancy of Pol II (new Fig. S7b).

These results suggest that knockdown of MED26 interferes with multiple transcription processes including transcription initiation, elongation, and termination. In contrast, it is likely that knockdown of ICE1 interferes with transcription termination, but not initiation and elongation, of RDH genes. Based on our results, we propose a model that (i) MED26 recruits LEC to the snRNA and RDH genes and plays a role in transcription processes including initiation and elongation, (ii) LEC subsequently binds to CBCA–NELF–DSIF and cooperatively inhibits read-through by Pol II at these genes, and (iii) LEC finally promotes 3'-processing of RDH genes and snRNA genes through recruitment of 3'-end processing factors for RDH genes or Integrator complex, respectively (new Fig. 10).

Taken together, it is understandable that the effect of MED26 knockdown was greater than that of ICE1 knockdown, because MED26 knockdown likely affects multiple transcription events including transcription initiation, LEC recruitment, Pol II elongation, CBCA-NELF assembly with LEC, and HLF recruitment around the transcription termination sites. This could be the reason why the transcription termination defect after MED26 knockdown was greater than that after ICE1 knockdown in the RNA-seq analyses shown in Fig. 2a and e. We have added relevant comments to the Results section and stated: “These results suggest that knockdown of MED26 interferes with multiple transcription processes including transcription initiation, elongation, and termination, while knockdown of ICE1 is likely to interfere with transcription termination only, consistent with the result of the polyA-selected RNA-seq showing that the transcription termination defect by MED26 knockdown was greater than that by ICE1 knockdown (Fig. 2a and e).” (Page 21, lines 3–8)

3) Since Med26 also recruits SEC, what is the effect of Med26 KD on read-through of an SEC dependent gene (which is a different question than the effect of AFF4 on an RDH gene).

As discussed earlier, we have added new data showing the effect of MED26 knockdown on read-through (UT/CT ratio) of the *c-myc* gene, a well-characterized SEC target gene, and the *GAPDH* gene. As shown in the revised Fig. 2a, depletion of MED26 did not affect the read-through transcripts at either the *c-myc* or *GAPDH* genes. (Page 10, lines 23–Page 11, lines 3)

In addition, we have added a new genome browser view of the *c-myc* gene in the PRO-seq analysis. As shown in the new Fig. 4g, read-through transcripts of the *c-myc* gene were not increased in MED26 hypomorphic mutant cells, indicating that N-terminal deletion of MED26 did not affect the read-through transcripts of SEC-dependent genes. (Page 13, lines 21–24)

4) KD of ZC3H8 also increases the UT/CT ratio, but except for H1H3E, the ratios are less than 0.1% . Even if statistically significant, what is the biological significance of such a small increase in polyA+histone mRNA? This needs to be discussed in terms of the overall biological significance of the LEC control of 3' end processing.

We appreciate the reviewer for raising this issue. Poly A-selected RNA-seq analysis

using ZC3H8 knockdown cells revealed that knockdown of ZC3H8 increased the polyadenylated RDH genes (new Fig. 2b and c), but the number of RDH genes affected by ZC3H8 knockdown was much less than that affected by knockdown of MED26 or ICE1.

Although ZC3H8 was reported to be a novel component of LEC, our results suggest that ZC3H8 is present in only a fraction of LECs. As shown in revised Fig. 5c, we found that F-ICE1 copurified lower amounts of ZC3H8 than LEC core components including ICE2, ELL, and EAF1. This result raises the possibility that ZC3H8 may act at only some loci or may be less important for LEC function than the more stoichiometric components of the complex. We have added relevant comments to the Results section and stated: “As expected, ICE1 copurified with other LEC components including large amounts of ICE2, ELL, and EAF1. It also copurified with smaller amounts of ELL2, EAF2, and ZC3H8, suggesting that (i) the majority of LEC in these cells is associated with ELL and EAF1 rather than ELL2 and EAF2 and (ii) ICE1, ICE2, and ELL/EAF1 are core components of LEC, while ZC3H8 is associated with only a subfraction of the LEC we have isolated (Fig. 5c). This observation, together with the fact that the number of genes associated with ZC3H8 was much greater than the number associated with ELL in ChIP-seq analyses (Fig. 3c), raises the possibility that ZC3H8 has functions outside of LEC.” (Page 15, lines 17–24)

The reviewer’s question about the overall biological significance of LEC control in 3’-end processing of RDH genes is an important one that will require further study. We would argue that even if the effect of MED26 or LEC subunit depletion on termination and processing is small, at others it is quite substantial – for example, ZC3H8 depletion increases the level of unprocessed HIST1H3E from ~10% - ~20% of total. It has been shown that non-polyadenylated Histone mRNAs are only stable during S-phase to prevent harmful production of free Histone proteins in cells outside of S-phase², because excess Histone levels lead to cytotoxicity through multiple mechanisms¹². It has also been shown that stem-loop binding protein (SLBP), which is only present during S-phase, binds to the conserved stem loop structure of RDH transcripts to prevent non-polyadenylated RDH mRNAs from undergoing degradation during S-phase². At the end of S-phase, SLBP is phosphorylated by cyclin-dependent kinase and degraded by the ubiquitin proteasome pathway. Degradation of SLBP outside of S-phase is tightly associated with disappearance of Histone mRNAs outside of S-phase³. Thus, SLBP plays an important role in ensuring high levels of Histone mRNAs and Histone proteins at S-phase to prevent harmful accumulation of free Histone proteins in cells outside of S-phase^{2, 4, 5, 6}. Considering that SLBP also

copurified with ICE1 (Revised Fig. 6a) and was identified as a component of CBCA-NELF-DSIF⁷, our results suggest that LEC and SLBP play a role in interfering with polyadenylation of RDH mRNAs to prevent harmful production of free Histone proteins in cells outside of S-phase. (Page 22, lines 12–21)

In support of this notion, the results of our plasmid-based assay suggest that the promoter of RDH genes is required for proper transcription termination as well as inhibition of Pol II read-through at RDH genes and aberrant Histone protein production. In our plasmid-based assay, we exchanged the HIST1H1C promoter with the Cytomegalovirus (CMV) promoter, a well-characterized viral promoter that supports transcription of polyadenylated transcripts. As shown in the new Fig. S8a, we generated two kinds of plasmids that contained the HIST1H1C promoter or CMV promoter, the HIST1H1C gene region from TSS to TES, and the region from TES to 100 bp downstream of TES that has a polyadenylation site (PAS). In addition, we added a FLAG-tag sequence at the C-terminus of the HIST1H1C gene to distinguish the protein from endogenous Histone H1. We transiently transfected the plasmids and compared the levels of total transcripts, read-through transcripts, and proteins. As shown in the new Fig. S8b and c, replacement of the HIST1H1C promoter with the CMV promoter increased the read-through transcripts, but not the total transcripts, suggesting that the HIST1H1C promoter is required for proper transcription termination and interferes with read-through of Pol II. This result is consistent with our notion that LEC recruited by MED26 to RDH genes interferes with read-through of Pol II at these genes. Intriguingly, replacement of the HIST1H1C promoter with the CMV promoter also increased the protein levels of HIST1H1C-FLAG (new Fig. S8d). This result is consistent with previous studies indicating that polyadenylation of Histone transcripts increases their stability and leads to increased levels of Histone proteins^{5, 6, 12}. These results indicate that the HIST1H1C promoter is required for proper transcription termination as well as inhibition of Pol II read-through and aberrant Histone protein production, consistent with our notion that LEC recruited by MED26-containing Mediator to the promoter of RDH genes interferes with read-through of Pol II at these genes. Taken together, our results suggest that MED26-containing Mediator recruits LEC to the promoter of RDH genes to interfere with polyadenylation of RDH mRNAs and inhibit aberrant production of free Histone proteins in cells outside of S-phase. (Page 21, lines 11–Page 22, lines 11)

To discuss this issue, we have added relevant comments to the Results section and stated: “Our results provided hints toward answering the important question of the overall biological significance of Mediator and LEC regulation in 3’-end processing of

RDH genes. Non-polyadenylated Histone mRNAs were shown to be regulated to be stable only during S-phase to prevent harmful production of free Histone proteins in cells outside of S-phase, because excess histone levels can lead to cytotoxicity through multiple mechanisms^{9, 47, 48, 49, 50}. Degradation of SLBP by the ubiquitin proteasome pathway outside of S-phase is associated with disappearance of Histone mRNAs outside of S-phase^{9, 51}. Considering that SLBP also copurified with ICE1 (Fig. 6a) and has been shown to associate with CBCA-NELF-DSIF²², our results suggest that LEC and SLBP could play a role in interfering with polyadenylation of RDH mRNAs to prevent inappropriate production of free Histone proteins in cells outside of S-phase.” (Page 22, lines 12–21)

To further address the biological significance of MED26-containing Mediator and LEC in transcriptional regulation of RDH genes, we next examined whether LEC recruitment by MED26 affects the induction of RDH genes at S phase^{2, 13} and cell cycle progression. Since the MED26 NTD is required for Mediator interaction with LEC and recruitment of LEC to the genes⁹, we took advantage of the MED26 hypomorphic mutant cells expressing a MED26 NTD deletion mutant. Using aphidicolin (APH) to arrest the cell cycle at S-phase, we examined the induction of RDH genes and observed

cell cycle progression after release of cells from S-phase by removing APH. As shown in Fig. S5c, induction of RDH genes after release of the cells from S-phase by APH removal was decreased in MED26 hypomorphic mutant cells, consistent with the ribo-depleted RNA-seq results showing that the majority of RDH transcripts were decreased in MED26 hypomorphic mutant cells (new Fig. S5a). Furthermore, as shown in panels (a) and (b), fluorescence-activated cell sorting (FACS) analysis revealed that the S-phase of MED26 hypomorphic mutant cells was prolonged and much longer than that of wild-type cells. Although the cell cycle defect observed in MED26 hypomorphic mutant cells could be due to a multitude of reasons, including not only changes expression of LEC-target genes, but also changes in expression SEC-target and other genes, it will be of interest in the future to explore the possibility that decreased RDH gene induction and alterations in 3'-processing shown in new Fig. S5c could contribute to the delayed release from S-phase observed in MED26 hypomorphic mutant cells.

5) In Fig. 2c,d the Venn diagram and table presumably summarize the status of polyA+ RDH RNA following KD. However, neither the figure nor the legend so indicates. If this is not the case, the results are inconsistent with previous results.

We thank the reviewer for raising this issue. The Venn diagram results in the original Fig. 2c and d were obtained by polyA-selected RNA-seq analysis. In accordance with the reviewer's suggestion, we have now stated "PolyA-selected mRNA" in the new Fig. 2d.

6) The majority of ICE1 regulated genes that show an increase in polyA+RNA do they not overlap with Med26. Why?

We thank the reviewer for raising this issue. There is a report that ICE1 functions in not only transcriptional regulation, but also other cellular functions including nonsense-mediated decay (NMD) of mRNAs¹⁴. Thus, ICE1 depletion could affect not only transcription of RDH and snRNA genes, but also NMD of mRNAs from other genes. This could be a reason why knockdown of ICE1 affected the expression of mRNAs that are not regulated by MED26. In the Results section of the revised manuscript, we state: "We note that many transcripts upregulated by ICE1 knockdown were not affected by MED26 knockdown. It was recently shown that ICE1 has a role in nonsense-mediated decay (NMD) of mRNAs outside the context of LEC³⁸; thus, it is possible that ICE1 knockdown affects not only transcription of RDH and snRNA genes but also NMD, and

perhaps other functions, at other genes. ” (Page 11, lines 11–15)

Our microscopic analyses provided data consistent with the possibility that ICE1 could have LEC-independent functions outside of the nucleus. In particular, when we quantified the number of ICE1 particles stained by anti-ICE1 antibodies, we found that 43 of 132 ICE1 particles in 33 cells were colocalized with coilin at Cajal bodies, and 89 ICE1 particles were not colocalized at Cajal bodies in nuclei, while 130 ICE1 particles in these cells were present in the extranuclear area and had smaller sizes than the particles colocalized with coilin in nuclei (new Fig. 7h). We have added the following sentences to the Results section: “Intriguingly, we observed the extranuclear regions of 33 cells and found that 130 ICE1 particles were present in the extranuclear area and that these particles had a much smaller size than particles colocalized with coilin in nuclei. Considering a recent report that ICE1 plays a role in NMD of mRNAs³⁸, it is possible that ICE1 has a role in NMD of mRNAs in the extranuclear region. ” (Page 19, lines 4–8)

Fig. 3 -

7) The Figure shows that published ChIP-seq data of ELL and ZC3H8 overlap with MED26 ChIP-seq, consistent with MED26 recruiting the LEC to RDH genes. However, the number of genes associated with ZC3H8 is much greater than with ELL. Since ELL is in both SEC and LEC, while ZC3H8 is only in LEC, why are the number of genes associated with ZC3H8 so much greater? Why isn't there a greater overlap of MED26 with ELL and ZC3H8? The authors need to discuss these points.

As the reviewer pointed out, the number of genes associated with ZC3H8 peaks was much greater than the number associated with ELL. Whether this is because ELL is really associated with a surprisingly small number of genes or because the available ELL antibodies are of relatively poor quality for ChIP remains to be determined. We think that what is more noteworthy is that snRNA and RDH genes are among the population of genes that indeed do show good overlap.

We also note that the ChIP-seq data for MED26 were obtained from HEK293T cells and we performed this analysis ourselves, while the ChIP-seq data for ELL and ZC3H8 were from published ChIP-seq analyses done by others and derived from HCT116 cells¹⁵. To compare the ChIP-seq peaks of MED26, ELL, and ZC3H8 using the same cell line, we performed ChIP-seq analysis of ELL and ZC3H8 using HEK293T cells, similar to our experiments for MED26 ChIP-seq. We found ChIP-seq peaks of

ELL in a subset of snRNA and RDH genes, and the results were similar to those obtained with the published data from HCT116 cells. However, it was difficult for us to carry out ChIP of ZC3H8 using the commercially available antibody (ab113260; Abcam). Thus, we had difficulty in comparing the ChIP-seq peaks of MED26, ELL, and ZC3H8 using HEK293T cells. As mentioned in our response to Comment #2) of Reviewer #2, levels of polyadenylated read-through transcripts of RDH genes were significantly increased by depletion of MED26 in both HCT116 and HEK293T cells (revised Fig. 1a–e). This result indicates that regulation of transcription termination by MED26 and LEC is at least conserved in different types of human cells. Taken together, it is convincing to compare the ChIP-seq data derived from HEK293T cells and HCT116 cells.

As mentioned in our response to Comment #4), ZC3H8 was reported as a novel component of LEC, although our result suggests that ZC3H8 is one of the subcomponents of LEC. As shown in revised Fig. 5c, we found that F-ICE1 copurified with a smaller amount of ZC3H8 than LEC core components including ICE2, ELL, and EAF1. This result raises the possibility that ZC3H8 is a subcomponent of LEC and it is possible that ZC3H8 has a different role from LEC in cells. We have added relevant comments to the Results section and stated: “As expected, ICE1 copurified with other LEC components including large amounts of ICE2, ELL, and EAF1. It also copurified with smaller amounts of ELL2, EAF2, and ZC3H8, suggesting that (i) the majority of LEC in these cells is associated with ELL and EAF1 rather than ELL2 and EAF2 and (ii) ICE1, ICE2, and ELL/EAF1 are core components of LEC, while ZC3H8 is associated with only a subfraction of the LEC we have isolated (Fig. 5c). This observation, together with the fact that the number of genes associated with ZC3H8 was much greater than the number associated with ELL in ChIP-seq analyses (Fig. 3c), raises the possibility that ZC3H8 has functions outside of LEC.” (Page 15, lines 17–24)

Fig. 4-

8) The authors conclude from Fig. 4 that the Med26 mutant enhances read-through. However, the metadata analysis in fig. 4D, clearly shows that the read counts in the gene body are also increased such that the overall distribution of reads does not appear to be different between WT and Mut in histone genes. This increase is clearly apparent in browser views in a) and c). The UT/CT ratio, based on number of reads, should be calculated from the metadata to determine whether there is a significant difference between WT and mutant Med26. The CT calculation should be based on reads downstream of the promoter, not upstream

(see below).

We thank the reviewer for raising this issue. In accordance with the reviewer's suggestion, we calculated a "PRO-seq read-through ratio". The "PRO-seq read-through ratio" was defined as "sum of reads from 500 bp to 1000 bp downstream of transcription end site (TES)" divided by "sum of reads from TES to 50 bp upstream of TES". As shown in the new Fig. S6e, the PRO-seq read-through ratios of RDH genes and snRNA genes in mutant (MUT) cells were significantly higher than those in wild-type (WT) cells. In contrast, the PRO-seq read-through ratios of other protein-coding genes were similar to those in WT cells. As shown in the new Fig. 4h, the PRO-seq read-through ratios in MUT cells divided by the PRO-seq read-through ratios in WT cells were significantly higher for RDH and snRNA genes than for other protein-coding genes. These results indicate that Pol II showed significant read-through of RDH and snRNA genes in MUT cells compared with WT cells. (Page 14, lines 1–8)

9) In the supplementary figure, a metadata analysis of UT/CT has been done. However, the ratio determined was comparing reads 100bp upstream to the downstream reads. This would not distinguish between increases of readthrough alone from increases across the gene body. The data should be recalculated comparing reads 50bp downstream of the promoter with readthrough.

We thank the reviewer for raising this issue. As mentioned in our response to Comment #8), we calculated the "PRO-seq read-through ratio". The "PRO-seq read-through ratio" was defined as "sum of reads from 500 bp to 1000 bp downstream of transcription end site (TES)" divided by "sum of reads from TES to 50 bp upstream of TES". As shown in left panel of new Fig. 4h, the PRO-seq read-through ratios of RDH genes and snRNA genes in mutant (MUT) cells were significantly higher than those in wild-type (WT) cells. In contrast, the PRO-seq read-through ratios of other protein-coding genes in MUT cells were similar to those in WT cells. (Page 14, lines 1–8)

In addition, we defined the "Pol II read-through ratio" as "sum of Pol II reads from TES to 1000 bp downstream of TES" divided by "sum of Pol II reads from transcription start site (TSS) to 1000 bp downstream of TES". As shown in right panel of new Fig. 4h, the Pol II read-through ratios of RDH genes and snRNA genes, but not other protein-coding genes, were significantly higher in MUT cells than in WT cells. (Page 14, lines 8–13)

10) snRNA genes more clearly show increased readthrough. However, the meta effects are modest. The analysis should be of polyA+ histones, as in Fig. 1.

Because it has been established that snRNA genes do not generally contain conserved cryptic polyadenylation sites (PAS), there are very few polyadenylated snRNA transcripts in cells even if Pol II read-through has occurred at snRNA genes¹⁶. Instead, to detect read-through snRNAs in cells, we took advantage of PRO-seq analysis using MED26 hypomorphic mutant (MUT) HEK293T cells and wild-type (WT) cells. As mentioned in our response to Comment #8), we calculated the “PRO-seq read-through ratio” defined as “sum of reads from 500 bp to 1000 bp downstream of transcription end site (TES)” divided by “sum of reads from TES to 50 bp upstream of TES”. As shown in left panel of the new Fig. 4h, the PRO-seq read-through ratios in MUT cells divided by the PRO-seq read-through ratios in WT cells were significantly higher for snRNA genes than for other protein-coding genes. These results indicate that there is a termination defect in MUT cells and that significant Pol II read-through occurs for snRNA genes in MUT cells. (Page 14, lines 1–8)

Fig. 5-

11) Both pull-down gel and mass spec analyses document that ICE1 is recovered in a complex with other LEC, CBCA-NELF and termination factors. However, it is surprising that Med26 is not recovered in either the pull-down gel or mass spec. This brings into question the entire premise of the study that Med26 recruits the LEC. The authors need to explain this.

We appreciate the reviewer for raising this important point. As mentioned in our response to Comment #4) of Reviewer #2, we agree with the reviewer and have added the results of new experiments showing that FLAG-tagged ICE1 (F-ICE1) copurifies with MED26-containing Mediator. We purified F-ICE1 from 293FRT cells stably expressing F-ICE1 and performed western blot analyses. As shown in the new Fig. 6b, F-ICE1 copurified with Mediator subunits including MED26, MED1, and MED23, while a control protein, F-BTBD19, did not. Of note, both F-ICE1 and F-AF9 copurified with Mediator subunits including MED26 and MED1 (Revised Fig. 6a), consistent with previous data showing that MED26 interacts with SEC^{8,9}. These results showed that ICE1 interacts with MED26-containing Mediator. As the reviewer pointed out, we did not detect MED26 in F-ICE1 immunoprecipitates in our proteomics analysis (Fig. 5b and revised Fig. 5c). Because protein modifications, especially

phosphorylation, interfere with ionization of peptides and identification of proteins by mass spectrometry, one possibility is that protein modifications such as phosphorylation make it difficult to detect MED26 by mass spectrometric analysis of F-ICE1-associated proteins.

12) NSAF is defined in the supplementary figure, but it should be defined in the legend to fig. 5.

We thank the reviewer for this comment. In accordance with the reviewer's suggestion, we have defined NSAF in the legend to Fig. 5 (page 53, lines 8–10) and in “MudPIT analysis” in the Methods section (Page 33, lines 15–Page 34, lines 18).

Fig. 8 –

13) The role of ICE1 in recruiting HLF and Integrator is assessed by determining the effect of ICE1 knock-down on CPSF73 and Ints9, respectively. However, neither of those proteins appears in the mass spec results shown in Fig. 5.

Therefore, the authors need to explain how those two proteins were chosen. Were others that did show up in the mass spec tested?

As shown in the revised Fig. 5c, CPSF3 (CPSF73) was detected as an ICE1-binding protein. In addition, western blot analysis revealed that both CPSF3 (CPSF73) and INTS9 copurified with F-ICE1 (Revised Fig. 6a). For this reason and because they are important components of HLF and INTS9, respectively, we chose these proteins for the ChIP assays. To address this issue, we have added relevant comments to the Results section and stated: “Because we found that the HLF component CPSF3 and Integrator component INTS9 copurified with FLAG-tagged ICE1 (Fig. 5c and Fig. 6a), we examined whether knockdown of ICE1 affected the occupancy of CPSF3 or INTS9 at RDH or snRNA genes, respectively.” (Page 20, lines 9–12).

14) In conclusion, the authors report an interesting study in which 3' processing factors and cap-binding factors are in a complex with the LEC, which is recruited to RDH and snRNA genes by Med26, resulting in proper 3' end processing. However, to firmly document their model, the authors need to address two issues:

1. Is the observed increased read-through with Med26 KD due to defective 3' end processing or to increases across the gene body which spill over to the 3' end?

We thank the reviewer for raising this issue. To address the reviewer's question, we calculated Pol II read-through ratios using Pol II ChIP-seq data from wild-type HEK293T (WT) cells and MED26 hypomorphic mutant HEK293T (MUT) cell lines. We defined "Pol II read-through ratio" as "sum of Pol II reads from TES to 1000 bp downstream of TES" divided by "sum of Pol II reads from transcription start site (TSS) to 1000 bp downstream of TES". As shown in the right panel of new Fig. 4h, the Pol II read-through ratios for both RDH genes and snRNA genes, but not other protein-coding genes, were significantly higher in MUT cells than in WT cells. This result indicates that the observed increases in read-through transcripts after MED26 knockdown or N-terminal deletion of MED26 were caused by defective 3'-end processing, and not by an increase in Pol II across the gene body that spills over to the downstream of TES (3'-end). (Page 14, lines 8–13)

2. What is the effect of replacing an RDH promoter with a promoter dependent on the SEC, to determine the effect on 3' processing.

We thank the reviewer for this suggestion. In general, it is technically difficult to replace the endogenous promoter of the HIST1H1C gene with the promoter of, for example, the *c-myc* gene, as one of the SEC target genes, in cells, even if we take advantage of the CRISPR system. Instead, we used a plasmid-based assay to examine the effect of promoter exchange in cells. We used the Cytomegalovirus (CMV) promoter, a well-characterized viral promoter that supports transcription of polyadenylated transcripts, because the *c-myc* gene promoter is much longer than the HIST1H1C promoter and it is not easy to define the minimum *c-myc* gene promoter region sufficient to regulate transcription.

As shown in the new Fig. S8a, we generated two kinds of plasmids that each contained the HIST1H1C promoter or CMV promoter, the HIST1H1C gene from TSS to TES, and the region from TES to 100 bp downstream of TES that had a polyadenylation site (PAS). The CMV promoter is generally used to express proteins of polyadenylated genes. In addition, we added a FLAG-tag sequence to the C-terminus of the HIST1H1C gene in plasmids to distinguish the protein from endogenous Histone H1. We transiently transfected the plasmids and compared the levels of total transcripts, read-through transcripts, and proteins. As shown in the new Fig. S8b and c, replacement of the HIST1H1C promoter with the CMV promoter increased the read-through transcripts, but not the total transcripts, suggesting that the HIST1H1C promoter is

required for proper transcription termination and interferes with read-through of Pol II. This result is consistent with our notion that LEC recruited by MED26 to RDH genes interferes with read-through of Pol II at these genes. In addition, replacement of the HIST1H1C promoter with the CMV promoter increased the protein levels of HIST1H1C-FLAG (new Fig. S8d). This result is consistent with previous reports indicating that polyadenylation of Histone transcripts increased their stability and led to increased levels of Histone proteins². Our results indicate that the HIST1H1C promoter is required for proper transcription termination as well as inhibition of Pol II read-through and aberrant Histone protein production, consistent with our notion that MED26-containing Mediator recruits LEC to the promoters of RDH genes to interfere with polyadenylation of RDH mRNAs and inhibit aberrant production of free Histone proteins in cells outside of S-phase. To discuss this issue, we have added relevant comments to the Discussion section and stated: “It has been shown that Histone mRNAs are only stable during S-phase, because SLBP, which is only present during S-phase, binds to the stem loop structure of RDH transcripts to prevent RDH mRNAs from undergoing degradation during S-phase⁹. Degradation of SLBP by the ubiquitin proteasome system outside of S-phase is associated with disappearance of Histone mRNAs outside of S-phase^{50, 51}. Furthermore, our plasmid-based assay revealed that replacement of a non-polyadenylated gene promoter (HIST1H1C promoter) with a polyadenylated gene promoter (CMV promoter) resulted in increased levels of read-through RDH transcripts and proteins. Considering that SLBP copurified with ICE1 (Fig. 6a) and was shown to be a component of CBCA-NELF-DSIF²², our results indicate that MED26-containing Mediator and LEC play a role in interfering with polyadenylation of RDH mRNAs to prevent harmful production of free Histone proteins in cells outside of S-phase^{46, 48}. ” (Page 27, lines 6–22)

Reviewer #4 (Remarks to the Author):

In this report, Takahashi et al reported that Med26 and the Little Elongation Complex (LEC) function in replication-dependent histone (RDH) and snRNA transcription termination. Mechanistically the authors provided evidence that the Med26 recruits LEC, which forms a large complex with various termination factors. Overall the study is well performed and presents potentially novel insights into RDH and snRNA termination as well as Mediator and LEC functions. However, there are some technical issues and their model is not fully substantiated by the data:

1) The authors proposed a model in which Med26 recruits LEC to RDH and snRNA genes. However, Flag-ICE1 immunoprecipitation failed to co-precipitate Med26 while Med31 was detected. The authors need to reconcile this with their model.

We appreciate the reviewer for raising this important point. As mentioned in our response to Comment #4) of Reviewer #2 and Comment #11) of Reviewer #3, we agree with the reviewer and have added the results of new experiments showing that FLAG-tagged ICE1 (F-ICE1) copurifies with MED26-containing Mediator. We purified F-ICE1 from 293FRT cells stably expressing F-ICE1 and performed western blot analyses. As shown in the new Fig. 6b, F-ICE1 copurified with Mediator subunits including MED26, MED1, and MED23, while F-BTBD19 did not. Of note, both F-ICE1 and F-AF9 copurified with Mediator subunits including MED26 and MED1 (Revised Fig. 6a), consistent with previous data showing that MED26 interacts with SEC^{8,9}. These results showed that ICE1 interacts with MED26-containing Mediator. As the reviewer pointed out, we did not detect MED26 in F-ICE1 immunoprecipitates in our proteomics analysis (Fig. 5b and revised Fig. 5c). Because protein modifications, especially phosphorylation, interfere with ionization of peptides and identification of proteins by mass spectrometry, our results raise the possibility that protein modifications such as phosphorylation make it difficult to detect MED26 by mass spectrometric analysis.

2) Instead of MED26 recruits heat labile factors to promote processing of RDH transcripts to form unpolyadenylated mRNAs, is it possible that MED26 inhibits the cleavage/polyadenylation pathway?

We appreciate the reviewer for raising this important point. As shown in revised Fig. 5c, FLAG-tagged ICE1 copurified with FLASH, which is a specific component of Heat labile factor (HLF) and is thought to be not present in the protein complex involved in the cleavage/polyadenylation pathway of polyadenylated genes¹⁷. This result suggests that LEC recruited by MED26 interacts with HLF and promotes 3'-processing of RDH genes, but might not inhibit the cleavage/polyadenylation pathway of other protein-coding genes. In addition, RT-PCR analyses indicate that cleavage at the *c-myc* and *GAPDH* genes is the same in cells expressing wild type MED26 or the MED26 hypomorphic mutant (Revised Fig. 2a). PRO-seq results also suggest that read-through

transcription at *c-myc* gene and numerous other protein-coding genes is not affected in MED26 hypomorphic mutant cells (new Fig. 4g and h). This result also supports our notion that MED26 does not inhibit the cleavage/polyadenylation pathway of normally polyadenylated transcripts.

3) Knockdown of ICE1 led to a decrease in the recruitment of CPSF73 and Ints9 (Fig. 8) and the authors argued that this is the evidence that LEC helps to recruit termination factors. Does Med26 knockdown also have the same effect?

We appreciate the reviewer raising this important point. To address the reviewer's question, we performed ChIP assays of CPSF3 (CPSF73) and INTS9 using HEK293T cells in which MED26 was knocked down. As shown in the new Fig. 9c and d, we found that knockdown of MED26 decreased the occupancy of CPSF3 (CPSF73) and INTS9 at RDH and snRNA genes, respectively. This result is consistent with our notion that LEC recruited to RDH or snRNA genes by MED26 plays a role in recruitment of HLF or Integrator to RDH or snRNA genes, respectively.

4) Does ICE1 knockdown lead to a decrease in Pol II density? If so, could the decrease in CPSF or Integrator recruitment be an indirect effect of lower Pol II activity?

We appreciate the reviewer for raising this important issue. To address the reviewer's question, we examined whether knockdown of ICE1 affects the occupancy of Pol II at the genes. As shown in the new Fig. S7b, ICE1 knockdown did not affect the occupancy of Pol II at RDH genes, indicating that the decrease in CPSF3 recruitment after knockdown of ICE1 is not an indirect effect of a decrease in Pol II (new Fig. 9a). In contrast, ICE1 knockdown moderately decreased the occupancy of Pol II at snRNA genes (new Fig. S7c). Considering that knockdown of ICE1 more substantially decreased the occupancy of INTS9 at snRNA genes (new Fig. 9b), our results indicate that the decrease in INTS9 recruitment to snRNA genes after ICE1 knockdown cannot be entirely explained by an indirect effect of a decrease of Pol II. Consistent with these results, knockdown of ICE1 moderately decreased the total transcripts of the U1 snRNA gene, but not RDH genes (new Fig. S7d). Taken together, our results support the notion that LEC contributes to the recruitment of 3'-end processing factors to RDH and snRNA genes.

5) In most studies, Med26 knockdown was performed. In assessing the effect on transcription, however, a hypomorphic mutant was used. What is the rationale for this? Is there a difference in the results between a knockdown and this mutant?

siRNA-mediated knockdown studies to investigate 3'-end processing were initiated in the Takahashi group, and the PRO-seq and Pol II ChIP-seq studies using the MED26 hypomorphic cell line were initiated in the Conaway lab as part of a different study. Upon realizing that we had obtained complementary results in the two systems, we decided to join forces. To address the reviewer's question whether results obtained in the two systems differ, we have performed additional RNA-seq and qPCR experiments to confirm (i) that polyadenylated RDH transcripts accumulate and (ii) that 3'-processing of RDH transcripts is defective in the MED26 hypomorphic cell line. We believe there is value to performing the experiments in the two systems (although we don't think it is necessary to perform ALL experiments in both), since the MED26 hypomorphic mutant cells generated from HEK293T cells express mutant MED26 that lacks the N-terminal domain (NTD) needed for the Mediator's interaction with LEC⁹, albeit at reduced levels relative to wild type MED26 in the parental cells.

References

1. Lyons SM, *et al.* A subset of replication-dependent histone mRNAs are expressed as polyadenylated RNAs in terminally differentiated tissues. *Nucleic acids research* **44**, 9190-9205 (2016).
2. Dominski Z, Marzluff WF. Formation of the 3' end of histone mRNA: getting closer to the end. *Gene* **396**, 373-390 (2007).
3. Zheng L, *et al.* Phosphorylation of stem-loop binding protein (SLBP) on two threonines triggers degradation of SLBP, the sole cell cycle-regulated factor required for regulation of histone mRNA processing, at the end of S phase. *Molecular and cellular biology* **23**, 1590-1601 (2003).
4. Brocato J, *et al.* Arsenic induces polyadenylation of canonical histone mRNA by down-regulating stem-loop-binding protein gene expression. *The Journal of biological chemistry* **289**, 31751-31764 (2014).
5. Costa M. Review of arsenic toxicity, speciation and polyadenylation of canonical histones. *Toxicology and applied pharmacology* **375**, 1-4 (2019).
6. Brocato J, Chen D, Liu J, Fang L, Jin C, Costa M. A Potential New Mechanism of Arsenic Carcinogenesis: Depletion of Stem-Loop Binding Protein and Increase in Polyadenylated Canonical Histone H3.1 mRNA. *Biological trace element research* **166**, 72-81 (2015).
7. Narita T, *et al.* NELF interacts with CBC and participates in 3' end processing of replication-dependent histone mRNAs. *Mol Cell* **26**, 349-365 (2007).
8. Takahashi H, *et al.* Human mediator subunit MED26 functions as a docking site for transcription elongation factors. *Cell* **146**, 92-104 (2011).
9. Takahashi H, *et al.* MED26 regulates the transcription of snRNA genes through the recruitment of little elongation complex. *Nature communications* **6**, 5941 (2015).

10. Florens L, Washburn MP. Proteomic analysis by multidimensional protein identification technology. *Methods in molecular biology* **328**, 159-175 (2006).
11. Washburn MP, Wolters D, Yates JR, 3rd. Large-scale analysis of the yeast proteome by multidimensional protein identification technology. *Nature biotechnology* **19**, 242-247 (2001).
12. Singh RK, Liang D, Gajjalaiahvari UR, Kabbaj MH, Paik J, Gunjan A. Excess histone levels mediate cytotoxicity via multiple mechanisms. *Cell cycle* **9**, 4236-4244 (2010).
13. Romeo V, Schumperli D. Cycling in the nucleus: regulation of RNA 3' processing and nuclear organization of replication-dependent histone genes. *Curr Opin Cell Biol* **40**, 23-31 (2016).
14. Baird TD, *et al.* ICE1 promotes the link between splicing and nonsense-mediated mRNA decay. *eLife* **7**, (2018).
15. Hu D, *et al.* The little elongation complex functions at initiation and elongation phases of snRNA gene transcription. *Mol Cell* **51**, 493-505 (2013).
16. Yamamoto J, *et al.* DSIF and NELF interact with Integrator to specify the correct post-transcriptional fate of snRNA genes. *Nature communications* **5**, 4263 (2014).
17. Marzluff WF, Koreski KP. Birth and Death of Histone mRNAs. *Trends in genetics : TIG* **33**, 745-759 (2017).

REVIEWERS' COMMENTS:

Reviewer #1 (Remarks to the Author):

The authors have replied to all my comments and modified the manuscript accordingly with the addition of new data and more quantitative data analysis. From my perspective, the revised version of the manuscript "The role of Mediator and Little Elongation Complex in transcription termination" is significantly improved.

I recommend acceptance of the manuscript.

Reviewer #4 (Remarks to the Author):

The authors have satisfactorily addressed my concerns.

REVIEWERS' COMMENTS:

Reviewer #1 (Remarks to the Author):

The authors have replied to all my comments and modified the manuscript accordingly with the addition of new data and more quantitative data analysis. From my perspective, the revised version of the manuscript “The role of Mediator and Little Elongation Complex in transcription termination” is significantly improved.

I recommend acceptance of the manuscript.

RESPONSE: We thank the reviewer for this comment.

Reviewer #4 (Remarks to the Author):

The authors have satisfactorily addressed my concerns.

RESPONSE: We thank the reviewer for this comment.